# Normal and Neoplastic Growth Suppression by the Extended Myc Network

**DOI:** 10.3390/cells11040747

**Published:** 2022-02-21

**Authors:** Edward V. Prochownik, Huabo Wang

**Affiliations:** 1Division of Hematology/Oncology, The Department of Pediatrics, UPMC Children’s Hospital of Pittsburgh, Pittsburgh, PA 15224, USA; huw14@pitt.edu; 2The Department of Microbiology and Molecular Genetics, The University of Pittsburgh School of Medicine, Pittsburgh, PA 15224, USA; 3The Hillman Cancer Center of UPMC, Pittsburgh, PA 15224, USA; 4The Pittsburgh Liver Research Center, Pittsburgh, PA 15224, USA

**Keywords:** ChREBP, L-Myc, Max, Mga, Mlx, Mnt, MondoA, Mxd, N-Myc, tumor suppressor

## Abstract

Among the first discovered and most prominent cellular oncogenes is *MYC,* which encodes a bHLH-ZIP transcription factor (Myc) that both activates and suppresses numerous genes involved in proliferation, energy production, metabolism and translation. Myc belongs to a small group of bHLH-ZIP transcriptional regulators (the Myc Network) that includes its obligate heterodimerization partner Max and six “Mxd proteins” (Mxd1–4, Mnt and Mga), each of which heterodimerizes with Max and largely opposes Myc’s functions. More recently, a second group of bHLH-ZIP proteins (the Mlx Network) has emerged that bears many parallels with the Myc Network. It is comprised of the Myc-like factors ChREBP and MondoA, which, in association with the Max-like member Mlx, regulate smaller and more functionally restricted repertoires of target genes, some of which are shared with Myc. Opposing ChREBP and MondoA are heterodimers comprised of Mlx and Mxd1, Mxd4 and Mnt, which also structurally and operationally link the two Networks. We discuss here the functions of these “Extended Myc Network” members, with particular emphasis on their roles in suppressing normal and neoplastic growth. These roles are complex due to the temporal- and tissue-restricted expression of Extended Myc Network proteins in normal cells, their regulation of both common and unique target genes and, in some cases, their functional redundancy.

## 1. Introduction

**The Myc and Mlx Networks.** The year 2021 marked the 30th anniversary of the discovery of Max, the obligate heterodimerization partner of the c-Myc (Myc) bHLH-ZIP transcription factor, which had itself been identified nearly a decade earlier as the evolutionarily conserved cellular homolog of the avian v-myc retroviral oncogene [1,2,3,4,5]. Shortly thereafter, it became clear that Max was necessary for Myc-mediated target gene activation and cellular transformation but that, at higher Max:Myc ratios, it also repressed these functions in a dose-dependent manner [6,7,8,9,10,11]. This initially suggested a simple model, whereby Myc–Max heterodimers, which contain a transactivation domain (TAD) contributed by Myc, and Max homodimers, which lack a TAD, alternatively activate or suppress transcription in accord with their relative abundance [12,13]. This balance was believed to be maintained by the competitive binding of these dimers to so-called “E-box” elements that are typically located in proximity to the transcriptional start sites of target genes (Figure 1). The model immediately implied that Max could serve dual functions—on the one hand, it could activate target genes and drive transformation by virtue of its obligate association with Myc; as a homodimer, on the other hand, it could outcompete Myc–Max binding, repress target gene expression and serve as a tumor suppressor (TS).

Max’s discovery soon paved the way for the rapid identification of six additional bHLH-ZIP Max partners that currently comprise the so-called “Myc Network”, of which Max remains the central member (Figure 1). These proteins, Mxd1–4, Mnt and Mga (hereafter referred to collectively as “Mxd proteins”, despite their structural and functional differences) (Figure 1), necessitated a revision of the above model as did the finding that, in mammalian cells, Max’s high level of phosphorylation, maintained by casein kinase II, prevents its binding to DNA as a homodimer but not as a heterodimer [19,20,43]. Transcriptional repression by Max was thus deemed to be mediated not simply by passive exclusion of Myc–Max heterodimers from binding E-boxes but as an active process that required Max’s heterodimerization with Mxd proteins, the association with obligate transcriptional co-repressors such as mSin3 proteins and the ensuing transcriptionally suppressive modification of chromatin via altered patterns of histone acetylation and methylation [14,28,44,45,46,47]. Collectively, these six Mxd proteins, which show distinct tissue-, developmental- and age-dependent patterns of expression, antagonize Myc’s broad impact on transcription and its highly pleiotropic effects on both normal and neoplastic growth [6,7,8,22,24,46,48,49,50,51,52,53,54,55].

By the turn of the 21st century, the existence of a second network, parallel to that of the Myc Network, and possessing significant structural and functional similarities, had emerged. This “Mlx Network”, also comprised of bHLH-ZIP factors, contained the Myc-like proteins ChREBP (“carbohydrate response element binding protein”) and MondoA and their own dedicated heterodimerization partner, the Max-related Mlx, which together formed the Network’s positively-acting arm [31,56,57,58] (Figure 1). Unlike Myc and Max, which are nuclear proteins, ChREBP, MondoA and Mlx are “conditionally nuclear” in that they translocate to the nucleus only upon binding metabolites such as glucose, glucose-6-phosphate, fructose 2,6-bisphosphate, lactate and adenosine [31,32,33,34,35,36,37,38,40,42,56,59,60,61,62,63,64,65,66,67,68]. The cytosolic location of MondoA and ChREBP is also not random; rather, amphipathic helical domains in the C-termini of these factors allow them to interact with intracytoplasmic lipid droplets and presumably serve as sensors of intracellular lipid content [69]. Lipid droplet depletion, together with the above metabolites, allows MondoA and ChREBP to translocate to the nucleus and activate genes involved in glucose metabolism. The negative arm of the Mlx Network shares three factors from the Myc Network, namely Mxd1, Mxd4 and Mnt, along with mSin3–histone deacetylase complexes [31,57,70]. These three negative regulators link the two Networks both structurally and functionally and provide a means of cross-talk for more refined and coordinated regulation of both common and unique targets.

Mlx Network heterodimers recognize E-box-related binding sites termed “carbohydrate response elements” (ChoREs) (Figure 1). Both E-boxes and ChoREs, but particularly the latter, can be quite variable and cross-binding of one Network’s members to the other Network’s sites is likely more common than previously appreciated [57,71,72,73,74,75,76]. Binding by ChREBP may also facilitate promiscuous, non-E-box-dependent binding by Myc at nearby sites [77]. Mlx Network target genes, while overlapping with those regulated by the Myc Network, are both fewer in number and have been reported to be more functionally restricted [39,41,78]. They tend to encode enzymes involved in carbohydrate and lipid metabolism; mitochondrial and ribosomal proteins and factors that regulate translational initiation, elongation and termination [31,39,41,42,46,70,71,72,73,76,78,79,80,81,82,83,84,85,86,87,88,89]. Together, these findings indicate that, via selective, cytoplasmic-nuclear partitioning, sharing of Mxd1,4 and Mnt and binding to both common and unique target genes, this “Extended Myc Network” cross-talks, while simultaneously communicating with and responding to the metabolic cues and reservoirs needed to support energy-intensive processes such as translation and proliferation [36,42,64,65,69,88,89,90,91,92,93,94,95,96].

**The Extended Myc Network and tumor suppression.** Given the number of Mxd proteins and their crucial roles, particularly with regard to limiting Myc signaling, proliferation and metabolism (Figure 1), it is reasonable to surmise that, at some level, they, and perhaps Max as well, behave as TSs. We discuss below the evidence to support this, the findings that the inactivation of these factors is often confined to certain tumor types and the implications of Mxd member haplo-insufficiency, which is a common theme in many tumors. It is also important to consider these factors in their proper biological context. In other words, do they behave as “classical” TSs such TP53, RB and BRCA1/2, whose germ line or somatic inactivation predispose to spontaneous neoplasms such as leukemias, lymphomas, retinoblastomas, osteosarcomas, and breast, ovarian and prostate cancer [97,98,99,100,101,102]? Or, does the inactivation of these factors simply accelerate the growth of pre-existing tumors without otherwise contributing directly to their initiation?

All Mxd members, as well as the central players Max and Mlx, have been implicated as TSs based on bioinformatics-based surveys of large populations of human tumors [103,104]. However, to our knowledge, neither comprehensive summaries nor intragroup comparisons of the consequences of copy number variations (CNVs) and mutations described in these reports have been published. By way of introduction to the more detailed discussions presented in the following sections, we surveyed all tumors from The Cancer Genome Atlas (TCGA) to identify those associated with CNVs of Extended Myc Network genes (Figure 2). This survey revealed several interesting findings. First, and in keeping with the known role of *MYC* as a bona fide oncogene and the presumptive roles of *CHREBP* and *MONDOA* in supporting transformation and rapid proliferation, these genes showed the highest frequency of amplification [36,41,45,103,104]. Second, recurrent deletions of all other Extended Myc Network members were quite frequent across many tumor types, in keeping with their presumptive role as TS genes (TSGs). Third, none of the genes was exclusively amplified or deleted across all tumor types or, in many cases, even within the same tumor type [104]. Fourth, among the “oncogenes”, high-level gene amplification, defined as >4.4 copies/cell [105], was seen only with *MYC* and tended to be associated with subgroups of certain cancers such as breast, ovary, liver and lung as previously reported [106,107,108,109]. Fifth, more often than not, amplifications or deletions of more than one Extended Myc Network gene were associated with particular tumor types and were most obvious in breast, ovarian and squamous cell lung cancers. This suggests that oncogenic activation of the Extended Myc Network can be achieved via multiple pathways that are cooperative rather than mutually exclusive. Sixth, despite its relatively high incidence of genomic instability, thyroid cancer was the only tumor in which CNV of any Extended Myc Network member was never observed (Figure 2). This may partly reflect the tumor’s high incidence of oncogenic *BRAF, PIK3CA* and *PTEN* mutations, which likely circumvent Myc Network activation [104,110]. In keeping with the theme that high-level *MYC* amplification tends to be confined to certain tumor types (Figure 2) [104], deletion of presumptive TSGs tend to cluster as well, particularly in those instances associated with two copy deletion as seen with *MLX* and *MNT* in ovarian cancer, *MXD2* in prostate cancer and *MXD3* in squamous cell lung cancer (Figure 2) [104]. Finally, the inactivation of TSG-like members of the Extended Myc Network by frame-shift and truncating mutations is rare, typically accounting for <1% of inactivated alleles, except in the case of *MGA*, where this approaches 4% [104]. Thus, as discussed more fully in the following sections, rather than the traditional bi-allelic inactivation/deletion of classical TSGs seen in both hereditary and sporadic cancers, hemizygous inactivation of *MXD* genes is not only common but a recurrent theme [98,99,100,101,102].

As was generally true for oncogenic members of the Extended Myc Network, the correlation between *MXD* TSG expression and survival is variable. Figure 3 summarizes these relationships among all tumors from TCGA and Figure 4 shows several specific examples of survival differences. Interestingly, survival correlated with the expression of as many as 8–9 of the individual Extended Myc Network members and in the cases of low-grade gliomas (LGGs) and clear cell renal cancers (KIRC), with all 11 members, although not necessarily in the same way. This supports the points mentioned above that perturbing the delicate balance among Extended Myc Network members may have oncogenic consequences that are additive or complementary rather than mutually exclusive. The connections among tumor aggressiveness, survival and the expression of individual members of the Extended Myc Network are thus complex, interdependent and presumably tumor type specific.

The relationships among Extended Myc Network gene CNVs, mutation, expression and survival are in many cases only correlative. Indeed, this cautionary note applies even to an oncogene as unassailable as *MYC* whose role in tumorigenesis mandated a direct demonstration of its actual causality [113,114,115]. Yet, recent work has indicated that, at least for some tumor types, Myc may function more as a facilitator of tumor growth than as an actual initiator. For example, hepatoblastoma (HB) in mice, induced via the ectopic expression of mutant forms of β-catenin and the Hippo pathway effector protein YAP, is associated with Myc over-expression [78,116,117]. Yet, the rate of tumor induction in livers with hepatocyte-specific knockout of Myc is identical to that seen in wild-type livers [41,78] although tumor growth is slowed. Similarly, tumors induced with different β-catenin mutants express widely different levels of Myc that roughly correlate with growth rates even though they too are initiated at identical frequencies [117]. These results, strongly indicate that the role of Myc in actual tumor induction versus facilitation is quite nuanced and may reflect both the levels at which it is expressed and the tissue environment [118].

A somewhat analogous situation, discussed in greater detail below, occurs with Max. While there is no evidence that *MAX* is an actual oncogene, it could be viewed as being another tumor facilitator that collaborates with Myc (Figure 1). Yet, the recurrent loss *of MAX* in pheochromocytomas, parganglioneuromas and several other tumors, as shown in Figure 2 and elsewhere, supports its role as a potent TSG [119,120,121,122,123,124,125]. Similarly, *MLX* loss dramatically impairs normal hepatocyte proliferation while at the same time serving as a potent suppressor of benign hepatic adenomatosis [76].

Given the well-accepted role of Extended Myc Network members in tumorigenesis, either directly as oncogenes or as tumor facilitators, we have focused here on their roles in tumor suppression and discuss each member in turn. As mentioned above and discussed in greater detail below, the functions of these members may differ, even in the same tissue. Where necessary, background information on the oncogenic function of these factors or their role in normal cell proliferation is provided as a way of furnishing context. We hope this review provides a much-needed source of information regarding this emerging function for the Extended Myc Network and how it impacts the balance between normal and neoplastic growth and the functions that support them.

## 2. Myc

Like *RAS* and *PI3K, MYC* is a classic example of a dominantly acting oncogene although, unlike the former two, it is seldom mutated in tumors. Rather, its oncogenicity commonly derives from gene amplification, where, along with *CCND1* and *EGFR*, it is among the top three amplified genes across a broad range of cancers (Figure 2) [105]. Such frequent, pronounced and unidirectional changes in gene copy number emphasize this pro-oncogenic role and are rivaled, albeit only modestly so, by similar changes in *CHREBP* (Figure 2). In other tumors, such as Burkitt’s lymphoma and subsets of diffuse large B-cell lymphoma (DLBCL) and multiple myeloma, *MYC*’s over-expression is a result of its translocation into immunoglobulin gene loci [126,127,128,129]. Yet these examples, involving alterations in chromosomal architecture, underestimate the true frequency with which MYC is dysregulated in cancer as aberrant signaling by many mutant growth factor pathways converge on *MYC* and promote its over-expression in the absence of structural changes [130,131,132]. However, as mentioned in the Introduction, it is important to distinguish between MYC’s role as a primary driver oncogene versus that of a tumor facilitator that simply provides metabolic and/or translational support without being necessary for tumor initiation [41,78,113].

Unlike the direct induction of genes mediated by Myc–Max DNA binding mentioned above (Figure 1), Myc’s role in transcriptional suppression is indirect and involves the association of Myc–Max heterodimers with transcription factors such as Miz1 that normally up-regulate genes encoding negative growth regulators such as *P15INK4B, P57KIP1* and *P21CIP1* [133,134,135,136,137]. Mechanistically, Myc’s bHLH-ZIP domain binds directly to Miz1 and prevents the latter’s binding to initiator (Inr) elements at transcription initiation sites of these target genes. Similarly, Sp1’s binding to multiple GGGCGG sites located in the *P21CIP1* promoter is disrupted via the interaction between its C-terminal zinc-finger and a ~200 residue internal region of Myc (amino acids 143–352) [138,139]. This effect is distinct from the suppression mediated by Miz1 since the deletion of *P21CIP1*’s Inr does not interfere with Myc’s blocking of SP1-mediated induction [138].

Despite its long and storied history as an oncoprotein, Myc may also play a role in tumor suppression. A novel means by which this might occur has been suggested by studies showing that ubiquitylation of several lysine residues in Myc’s C-terminus by the E3 ubiquitin ligase HectH9 is necessary for Myc to activate fully its target gene repertoire [140], possibly by licensing the latter protein’s interaction with and subsequent stabilization by the histone acetyltransferase (HAT) p300 [141]. HectH9 is over-expressed in many tumors and is inhibited by Miz1 [140]. This suggests that HectH9 is required for Myc to achieve its maximal transcriptional activation potential, presumably by optimizing its recruitment of p300 and other co-activators [16]. HectH9 thus represents a potential therapeutic target that could suppress if not entirely eliminate Myc’s impact on tumor growth by dictating its potency as an oncoprotein [140]. This further suggests that Myc toggles between two states of high and low transcriptional activation potential dictated by its degree of HectH9-mediated ubiquitination. Though perhaps not representing a true tumor suppressor in this latter context, the under ubiquitinated form of Myc could potentially behave in a dominant negative-like manner, thereby actively limiting tumor growth. Functionally, this could be viewed as being akin to one of the roles of Mxd proteins which is to functionally “inactivate” Myc by preventing its DNA binding (Figure 1).

Another post-translational modification that may contribute to Myc’s role in tumor suppression involves its direct phosphorylation at Thr358, S373 and Thr400 by the Pak2 Ser/Thr kinase, which inhibits Myc–Max heterodimerization and reduces target gene affinity [142]. In promyelocytic leukemia (PML) cells, this has an additional impact on retinoic acid (RA)-induced differentiation mediated by the RA receptor (RARα) and its induction of genes that inhibit proliferation and promote differentiation [143,144]. Unphosphorylated and phosphorylated Myc differentially contact RARα, with the former blocking RARα‘s transcriptional program and maintaining the proliferation of undifferentiated cells [145]. Unphosphorylated Myc’s binding to RARα decreases the latter’s association with co-activators while increasing its association with co-repressors, whereas the reverse occurs in response to Myc phosphorylation. Among the more prominent of these factors are the co-repressor HDAC3 histone deacetylase and the co-activator HAT CBP. The relationship between Myc and Pak2 is reciprocal in that Pak2 is up-regulated during late-stage RA-induced terminal PML differentiation when Myc levels are declining [146]. Thus, different environmental cues can allow Myc to either block or facilitate PML differentiation and proliferation. The mechanism by which Myc blocks PML differentiation, i.e., by directly interacting with and inhibiting another transcription factor, is formally analogous to the previous described inhibition by Myc of Miz1- and Sp1-mediated transcription.

Pak2-mediated phosphorylation of Myc has also been described in many other normal and neoplastic contexts, including primary hematopoietic cells, keratinocytes, fibroblasts and other leukemias [142,143,144,147,148,149]. It has been proposed that phosphorylated Myc plays a role in maintaining hematopoietic balance by promoting stem cell adherence and limiting expansion. However, the amplification of PAK2 (as well as PAK2-related genes) is considerably more common in cancer than is inactivation and missense mutations of Pak2 phosphorylation sites in Myc are virtually non-existent [143,144,150,151]. Thus, any role for Myc as a TS in primary human cancers is likely to be quite limited. The rare instances in which Myc seems to act as a “TS” are in actuality those in which post-translational modifications (or lack thereof) alter its stability or function in ways that reduce its effectiveness as an oncoprotein.

## 3. Max

Given Max’s central role in balancing the transcriptional output of the Myc Network (Figure 1), it might be perceived as promoting either pro-oncogenic or TS functions in a manner that depends on the relative abundance of Myc–Max heterodimers on the one hand and Mxd–Max heterodimers on the other [10,11,46,152]. Myc–Max association is needed to maintain both normal and tumor cell proliferation and inhibiting this for therapeutic purposes in cancer, whether by pharmacologic or genetic means, has been a long-desired goal [6,7,8,50,153,154,155,156,157,158]. Thus, disproportionate focus has been placed on Max in the context of its function as an obligatory pro-oncogenic co-factor for Myc-mediated transformation.

The first evidence for Max’s potential as a TS was provided by studies showing that, when co-expressed in relative excess to Myc, it suppressed Myc-mediated target gene activation and/or transformation in vitro [7,8,10,11,159,160]. Evidence for a natural TS function of Max was subsequently suggested by the observation that the PC12 pheochromocytoma (PC) cell line expressed a mutant Max transcript and no functional protein and that the enforced re-expression of wild-type Max inhibited PC12 growth [161].

Adrenal gland PCs and their extra-adrenal gland counterparts, paraganglioneuromas (PGLs), are typically benign tumors, approximately one-third of which are inherited and harbor identifiable germ-line mutations [162]. Additionally, approximately two-thirds of sporadic cases have an underlying genetic cause [119,163] involving at least 14 genes [120,121,122,125,164]. Max mutations are found in ~1–2% of tumors, particularly those which are inherited, display malignant features, arise in younger individuals and/or are associated with higher levels of metanephrine or normetanephrine secretion [120,122,162].

In the largest single study performed to date involving nearly 1700 inherited and sporadic PCs and PGLs, 20 Max mutations were identified, 18 of which were novel [120]. Nine of these were associated with mutations of the methionine initiation codon, with the remainder causing premature termination, aberrant splicing or in-frame deletion of crucial amino acids. The majority of these failed to express Max protein. The remaining 11 tumors contained missense mutations, seven of which were predicted to be deleterious, although all expressed the mutant Max proteins. Loss of heterozygosity (LOH) was observed in 16 of the 18 tumors analyzed.

MAX mutations have subsequently been described in tumors as diverse as multiple myeloma, small-cell lung cancer (SCLC), pituitary adenoma and quadruple wild-type gastrointestinal stromal tumor [165,166,167,168]. Multiple myeloma, in which the MAX mutation rate is ~3%, was a particularly informative source as the Max mutations provided clues into how they impacted the protein’s binding to various epigenetically-modified E-boxes [168]. The fact that methylation at position 5 of the internal cytosine residue (5mC) of the canonical CACGTG E-box inhibits Max binding has been known since the protein’s original discovery [169]. However, the E-box is subject to additional naturally-occurring cytosine modifications mediated by members of the ten-eleven translocation (Tet) family of Fe(II)- and α-ketoglutarate-dependent dioxygenase family, including 5-hydroxymethylation (5hmC), 5-formylation (5fC), and 5-carboxylation (5caC). All these modifications except 5caC have been shown to abrogate Max binding, particularly when the modification is present on both strands of the palindromic sequence [168]. A determination of Max’s crystal structure in homodimeric association with a 5caC-modified E-box showed that the R36 residue demonstrated the largest conformational difference when compared to the structure of Max or Myc–Max bound to the unmodified E-box. The importance of this residue was demonstrated by showing that the myeloma-associated Max mutation R36W completely abolished DNA binding. Moreover, of the 25 unique Max mutations identified among 805 multiple myelomas, 17 involved missense mutations, with five of these directly altering R36 and two altering the adjacent R35 residue. Finally, of the 643 tumor samples for which both mutational status and RNAseq results could be assessed, those harboring Max mutations expressed significantly lower levels of Myc and tended to display more favorable outcomes. The lower-level expression of Myc in these samples was likely due to the loss of its Max-mediated stabilization [168,170]. To explain how Myc drives myelomagenesis in the absence of efficient Max association, the authors suggested that Myc interacts with other E-box-binding proteins such as WDR5 [171,172].

SCLC in humans is often associated with *TP53* and *RB1* loss/mutation and amplification of *MYC* or its paralogs *MYCN* and *MYCL* [173,174,175]. A recent CRISPR/Cas9-based screen for suppressors of early-stage SCLC found Max to be among the top hits with no enrichment being observed for other members of the Extended Myc Network [176]. Confirmatory studies showed that Max knockdown in these “preSC” cells led to more robust growth and survival during both anchorage-dependent and independent growth in soft agar. When tested in vivo in an autochthonous *Rb1/Trp53*-deleted mouse model of SCLC [177], Max knockout was associated with significantly larger numbers of tumors and shortened survival. Max re-expression in a cell line derived from these tumor cells significantly suppressed growth. Interestingly, when preSC cells were engineered to over-express Myc, MycN or MycL, the concurrent knockout of Max inhibited growth. These findings suggested that, at least in this model system, Max’s role in SCLC development is context dependent and at least partly reliant upon the level of expression of Myc or its paralogs, particularly MYCL. While Max appeared to serve a pro-oncogenic function by facilitating *MYC* family-mediated transformation, it could also serve as a TS via one or more Myc-independent pathways.

RNAseq was performed in control preSC cells and SCLCs with or without MAX knockout as well as in SCLCs with MAX knockout and restored Max expression. Common deregulated genes shared by these cohorts included 113 that were up-regulated in response to MAX knockout and 56 that were down-regulated. Among the former were genes previously shown by ChIP to bind Myc–Max and Mxd–Max heterodimers. Collectively, these findings were consistent with the idea that MAX knockout reverses the suppression of genes mediated by the inhibitory arm of the Myc Network (Figure 1). Further analyses of these 113 genes and several hundred additional ones identified using less stringent criteria showed an enrichment for pathways dedicated to one-carbon metabolism along with the metabolism of nucleotide sugars, serine, alanine, aspartate and glutamate. Many of these genes’ promoters were co-occupied by Myc–Max and Mxd–Max heterodimers thus likely representing sites that could bind either heterodimer, depending on the cells’ metabolic and/or proliferative state. It was speculated that up-regulation of these genes in response to MAX knockout was due to the activity of other transcription factors that are otherwise normally impeded by Max–Mxd occupancy of nearby E-boxes. It was further surmised that the over-expression of *MYC* and its paralogs would displace Max–Mxd heterodimers from these sites, leading to increased Myc–Max binding and even more pronounced target gene induction.

The idea that Max plays a role in TS while also promoting transformation via its association with Myc suggests two simple and non-mutually exclusive ways by which this could occur. First, in some contexts, the loss of transcriptional balance between the positive and negative arms of the Myc Network may favor transformation due to distinct sets of genes whose expression is now altered (Figure 1). In other contexts, these target genes, no longer bound by any Myc Network components, are now excessively bound by other E-box-binding proteins and altered in ways the favor transformation.

## 4. Mxd1

Mxd1 (previously Mad1) was first identified as a Max partner and subsequently shown to possess the dimerization specificities depicted in Figure 1 [45,46,70,152]. In the mouse embryo Mxd1 mRNA is selectively expressed in a developmentally-dependent manner in tissues such as liver, epidermis and the brain’s mantle layer [178,179]. Postnatally, its expression broadens and becomes most prominent in differentiated tissues such as epidermal ketatinocytes and the apical and luminal regions of intestinal crypts [178,179]. These findings, along with the observation that Myc expression is more restricted to highly proliferative cellular compartments, suggested that Myc and Mxd1 were engaged in opposing functions. They thus contributed to the then nascent model that terminal differentiation and proliferative quiescence were driven by the dissolution of Myc–Max heterodimers and the de novo formation of Max–Mxd1 heterodimers. However, the model was complicated by studies showing that Myc–Max heterodimers and Max–Mad heterodimers engaged distinct subsets of E-box-regulated genes, albeit with some overlap [180].

Mxd1’s function as a TS in vivo was initially tested in a study that knocked out the gene in mice and found there to be no significant increase in spontaneous cancer incidence or other major phenotypes, thereby implying that other members of the Mxd family (Figure 1) were functionally redundant [48,49]. More careful inspection, however, reveled that myeloid cell maturation was impaired due to delayed cell cycle exit during in vitro differentiation in response to GM-CSF or G-CSF. It was postulated that, during the latter stages of differentiation, committed colony-forming granulocyte precursors normally expand in a Myc-dependent manner to a degree that is limited by the gradual expression of Mxd1 and diminution of Myc, ultimately giving rise to so-called cluster-forming cells of more limited proliferative capacity [48]. *Mxd* gene knockout allowed for the accumulation of these cells, which were also more dependent on G/GM-CSF for survival. Notably, no expansion of less mature progenitors was observed, which would have been expected with a more leukemogenic insult. This suggested that Mxd1 was more involved in the advanced stages of terminal differentiation as had been previously surmised from the studies discussed above. This was supported by the observation that *Mxd1* knockout bone marrow was modestly hypercellular due to an expansion of mature granulocytes [48,178,179,180]. Consistent with this, recovery from 5-flurouracil-induced bone marrow hypoplasia occurred more rapidly in Mxd1 knockout mice.

Other hints that Mxd1 might suppress normal or neoplastic under different circumstances came from studies showing that its enforced expression repressed Myc+ Ras-mediated transformation of primary rat fibroblasts [178,179] and impaired the in vitro growth of established human cancers in a manner that is consistent with the model depicted in Figure 1 [181,182,183]. However, subsequent studies in human cancers showed that Myc and Mxd1 were often co-expressed and poorly correlated [181,184,185]. Sustained proliferation in the face of relatively high levels of Mxd1 might be attributable to its post-translation inactivation. Indeed, the Ser/Thr kinases Akt, RSK and S6K, which are frequently over-expressed in cancer can directly phosphorylate Mxd1 at S145 and abrogate transcriptional repression by inhibiting DNA binding without disrupting its association with Max [184,186,187,188,189,190]. Akt-mediated phosphorylation did not alter Mxd1’s already short half-life, whereas RSK and S6K did, as evidenced by increased ubiquitination and proteasome-mediated proteolysis [184,190]. Akt-mediated phosphorylation did, however, negate DNA binding, thus shifting the gene expression balance to one that was primarily driven by Myc–Max heterodimers and Myc target gene activation. The distinct outcomes mediated by Akt, RSK and S6K on Mxd1 half-life may have been attributable to differences in the cell lines utilized to study these activities, to alternative Mxd1 phosphorylation sites and/or to other proteins that differentially interacted with the kinases to impact Mxd1 stability. Together, the findings indicated that high levels of mitotic signaling mediated by the PI3-kinase/Akt/S6K and MAP kinase/RSK pathways converge on Mxd1 so as to fine-tune its stability, its association with Max and its DNA binding activity, thus allowing for competing Myc-mediated cell cycle-promoting processes to proceed.

An alternative form of intranuclear regulation for Mxd1 and other Myc Network members was investigated by confocal microscopy in three different cell lines expressing GFP variant-tagged Myc, Max and Mxd1–3 [191]. Max expression alone was diffuse and homogeneous, whereas Myc and Mxd1–3 formed multiple discrete nuclear “speckles” that co-localized. These were dynamic and co-dominant in that, when co-expressed with Max, they assumed the pattern of whichever member of the pair was more abundant.

Co-localized Myc and Mxd3 nuclear speckles lacked even random overlap with those formed by the SC-35 splicing factor [148,191], thus suggesting that their separation was maintained by active processes. However, when GFP-Myc and untagged Max were expressed at near equal ratios that allowed the former protein to tenuously retain its speckles (presumably in association with Max), these now co-localized with ~15% of SC-35-containing speckles, particularly at the latter’s peripheries where the most active transcription is believed to occur [192,193,194]. In contrast, GFP-Mxd3 + Max speckles did not co-localize with SC-35 speckles. Thus, prior to dimerization with Max, Myc and Mxd3 (and probably Mxd1, Mxd2, and Mxd4 as well) appear to comprise a common population of transcriptionally inert subnuclear speckles. Myc–Max and Max–Mxd heterodimers are reapportioned into distinct domains with the former residing at sites of mRNA transcription and processing and the latter occupying more transcriptionally quiescent regions.

Among the most critical and highly coordinated Myc targets are genes which are regulated by all three RNA polymerases and encode ribosomal structural proteins, rRNAs, tRNAs and the translation initiation, elongation and termination factors needed to ensure maximal protein synthesis and tumor growth [41,78,113,195,196]. In addition to localizing to the nucleus as described above, Mxd1 may play a specific role in regulating rRNA synthesis by also localizing to the nucleolus [197]. The nucleolus’ fibrillar center (FC) is the site where RNA Pol I and Upstream Binding Factor (UBF) are concentrated [198]. FCs are embedded within the dense fibrillar component (DFC) that sequesters the ribosomal genes and harbors various RNA-binding proteins. Both FCs and DFCs represent sites of active Pol I transcription with subsequent pre-rRNA processing occurring within the DFC to generate mature 18S and 28S species [199]. The subsequent assembly of these into 40S and 60S ribosome subunits occurs within the granular component (GC) [200,201]. It was shown in several cell lines and primary tissues that nucleolar Mxd1 associates with UBF and becomes particularly prominent following serum starvation [197]. This is in sharp contrast to Myc and Mnt, both of which displayed nuclear segregation. Treatment of cells or whole animals with actinomycin D at low concentrations that selectively inhibited Pol I [202] led to a loss of nucleolar Mxd1 without affecting its nuclear compartmentalization. As had been previously demonstrated for Myc–Max complexes, chromatin immunoprecipitation studies verified that Mxd1 binding occurred throughout rDNA repeats often, but not always, in association with UBF [196,203].

In addition to the above-discussed post-translational varieties of regulation, Mxd1 stability is also regulated at the post-transcriptional level by miR19a/b, which are products of the six-member miR-17-92 cluster that is both over-expressed by and drives numerous cancers, promotes epithelial-mesenchymal transition and is a direct Myc target [204,205,206,207,208,209,210,211,212,213,214]. Individuals with gastric cancers which over-express miR-19a have significantly shorter survival and a higher incidence of metastatic disease at diagnosis than those with low expression [215]. Two gastric cancer cell lines engineered to over-express miR-19a or miR-19b, and displaying lower levels of Mxd1 as a consequence, demonstrated enhanced in vitro migration and invasion, whereas these behaviors were impaired when endogenous miR-19a/b were inhibited and normal Mxd1 levels were restored. In gastric cancer xenografts, miR-19a or miR-19b over-expression also increased both the frequency of metastatic deposits and their sizes. Additional experiments identified two direct miR-19a/b binding sites within the Mxd1 mRNA 3′-untranslated region. Finally, in addition to being positively regulated by Myc, the miR-17-92 cluster was also negatively regulated by Mxd1 over-expression, thus establishing a self-regulatory feedback loop among the *MYC*, *MXD1* and miR-19-92 loci [216].

Taken together, the above findings suggest that Mxd1 is transcriptionally regulated in ways that are highly tissue- and differentiation stage-dependent. Post-transcriptional regulation may be achieved by non-mutually exclusive means that include, but are not limited to, competition for available Max, selective sub-cellular partitioning and patterning and the differential regulation of the genes within these compartments such as those encoding rRNAs in the nucleolus. The basal expression of Mxd1 transcripts may be determined by the abundance of competing miRNAs such as miR-19a/b and MiR-17-92.

## 5. Mxd2

Initially isolated using a yeast two-hybrid screen that employed Max as a “bait”, Mxd2 (originally known as Mxi1) is expressed in a variety of tissues and up-regulated in response to the differentiation of U937 monocytic leukemia cells [217]. The human *MXD2* gene, located on chromosome 10q25.2 (https://omim.org/entry/600020, accessed on 3 July 2021), resides within a locus that is often subject to recurrent deletion and LOH in several human tumors, most notably over half of prostate cancers and glioblastomas [218,219,220,221,222,223,224,225,226,227,228,229,230,231]. Inactivating point mutations in the retained *MXD2* allele were initially identified in a rare subgroup of prostate cancers with cytogenetically detecTable 10q24-q25 deletions and subsequently in a larger and more common group of tumors harboring much smaller 10q25.2 deletions detectable only by fluorescence in situ hybridization [228,232]. Regardless of cytogenetic background, *MXD2* point mutations in the retained allele, both in prostate and other cancers, are infrequent and their subsequent confirmation has been inconsistent [233,234,235,236,237]. Importantly, the searches for these mutations, which were conducted prior to the employment of whole exome sequencing, typically relied upon insensitive detection methods such as single-strand conformation polymorphism screening that likely would not have detected mutations in small clonal subpopulations of tumor cells and/or in tumors with significant amounts of cytogenetically normal stroma that is a feature of most prostate cancers [221,234,235,238]. On the other hand, *Mxd2* point mutations were relatively common among a diverse group of 24 primary rat cancers and cell lines induced by a variety of known carcinogens [239]. In total, 6 tumors with mis-sense mutations and one with a frame-shift were identified across the protein. The consequences of the missense mutations on Mxd2 function were neither immediately obvious nor tested further. Taken together, the relative infrequency with which bi-allelic loss/inactivation of *MXD2* occurs in human cancers suggests that haplo-insufficiency, an increasingly recognized phenomenon in a number of cancers [240,241], may be more responsible for contributing to tumorigenesis.

A more convincing role for *Mxd2* as a TSG was obtained in a longitudinal study of engineered *Mxd2*^−/−^ mice [242]. These animals showed a propensity to develop splenomegaly in association with a pronounced expansion of splenic white pulp, extramedullary hematopoiesis and the occasional evolution to B-cell lymphomas within the first year of life. Mitotic stimulation of *Mxd2*^−/−^ splenic T cells with anti-CD3/CD28 antibodies led to significantly higher rates of proliferation and expansion relative to wild-type cells that correlated with a more efficient G0/G1 → S-phase entry. Together with the postulated role discussed above for Mxd1’s support of myeloid precursor proliferation, these findings suggest a broader role for these two negative factors in overseeing hematopoietic cell proliferation. Surprisingly, the authors were unable to demonstrate any proliferative abnormalities in *Mxd2*^−/−^ B cells in response to lipopolysaccharide-mediated proliferation, despite the fact that the malignancies originating in some of the mice were of B-cell origin.

Notable additional abnormalities in aging *Mxd2*^−/−^ mice involved degenerative changes in the renal cortex that were preceded by the cytoplasmic vacuolization of cells comprising the proximal tubular epithelium and focal atrophic degenerative changes in proximal tubules and glomeruli [242]. Some of the changes were reminiscent of autosomal dominant polycystic kidney disease (ADPCKD), which has been linked to Myc over-expression, is associated with renal epithelial hyperproliferation and can be generated by transgenic Myc deregulation [243]. These results were subsequently confirmed by DNA microarray expression profiling that compared renal epithelia from normal and *Mxd2*^−/−^ polycystic murine kidneys [244]. This study found fewer than 150 gene expression differences, including a ca. two-fold up-regulation of Myc and a variety of transcripts that participate in pathways pertaining to inflammation, the immune response and chemokine signaling [244]. The latter included significant up-regulation of the CKL12, CKL14 and CKL19 chemokines. These are related to IL-8, which is known to participate in ADPCKD pathogenesis [245,246]. Indeed, *Mxd2*^−/−^ human embryonic kidney cells significantly increased IL-8 secretion and activated the p38 MAPK pathway. It was speculated that *Mxd2* knockdown either directly or indirectly activated this pathway, which in turn lead to a pro-inflammatory phenotype that, in addition to increased secretion of IL-8, included *Myc* up-regulation and the ensuing uncontrolled renal epithelial proliferation.

The basis for ADPCKD was further investigated by establishing 3D in vitro cultures of murine inner medullary collecting duct (mIMCD-3) cells, which undergo branching tubulogenesis when exposed to epidermal growth factor and hepatocyte growth factor [247,248]. In response to these, tubulogenesis was reduced by nearly 10-fold in Mxd2-over-expressing mIMCD-3 cells but was restored when *Mxd2* was silenced. Genes encoding fibronectin, integrin and Mmp9, which are induced during tubologenesis and play key roles in this process [249,250], failed to be up-regulated in response to *Mxd2* knockdown. Moreover, the knockdown of *Mmp9* alone was sufficient to inhibit tubulogenesis in cells that expressed otherwise normal levels of Mxd2 thus establishing a pathway by which the enforced expression of Mxd2 inhibited tubulogenesis by preventing *Mmp9* induction. Together, these results suggest that Mxd2 tempers Myc’s tendency to drive an over-proliferation of medullary collecting duct cells and that both genes are needed for normal and properly balanced renal epithelial tubulogenesis.

Because of the previously discussed link between *MXD2* loss/mutation and human prostate cancer [228,232], Schreiber-Agus et al. also examined the prostates of older *Mxd2*^−/−^ male mice [242]. While not progressing to frank neoplasia, they did develop hypercellular acini and dysplastic cells with higher proliferative indices as determined by Ki-67 immunostaining. *Mxd2*^−/−^ MEFs also replicated faster in vitro than their wild-type counterparts and were 3–5-fold more transformable by *MYC+RAS* oncogenes. Collectively, these findings suggested that *Mxd2* gene silencing was associated with broadly similar outcomes in prostatic epithelium, MEFs and T cells that involved a dysregulation of both Myc and proliferative signaling. The susceptibility of *Mxd*^−/−^ MEFs to *MYC+RAS* transformation was likely due to Myc’s failure to be transcriptionally challenged by Mxd2, thus more efficiently suppressing Ras-induced senescence [251,252].

In addition to the modestly higher incidence of spontaneous lymphomas in *Mxd2*^−/−^ mice mentioned above, a more impressive increase in lymphomas and cutaneous squamous cell carcinoma was observed following DMBA treatment [242]. Finally, *Mxd2*^−/−^ mice crossed with cancer-susceptible *Ink4a*^−/−^ mice developed lymphomas and fibrosarcomas at a significantly higher rate and survived for a shorter time than did an *Ink4a*^−/−^ control group. Together, these findings converge on the idea that Mxd2 exerts broad anti-proliferative effects on a variety of cell types and that its loss contributes to the evolution of various cancers, particularly in cases where predisposing conditions already exist. Loss of competition between Myc–Max and Max–Mxd2 heterodimers may, however, only partially explain the cancer susceptibility of these mice since the *Myc* gene itself is negatively regulated by Mxd2 [253]. Cancer propensities in *Mxd2*^−/−^ mice may thus reflect both a greater abundance of Myc protein, more efficient Myc binding to its target genes, increases in the expression of target genes that normally bind Myc with low affinity and a loss of repression of key Mxd2 targets.

Work showing that ectopic Mxd2 expression induced G2/M growth arrest and suppressed the proliferation and/or colony-forming capacity of both prostate cancer and glioblastoma cells was broadly compatible with the above model [231,254]. Interestingly, the growth arrest of U-87 glioblastoma cells coincided with the formation of Max–Mxd2 heterodimers that bound to a consensus E-box in the proximal promoter of the *CCNB1* (cyclin B1) gene and suppressed its transcription [255]. This prompted a G2/M arrest due to an inability to accumulate sufficient cyclin B1 to complete mitosis. These findings were compatible with earlier work performed in murine 32D myeloid cells showing that the combination of Myc over-expression and TP53 inhibition promoted the spontaneous accumulation of tetraploidy that was markedly enhanced by sublethal doses of γ-irradiation and was accompanied by increased cyclin B1 and its associated cdc2 kinase activity [256]. It was shown that the above-mentioned E-box in the CCNB1 promoter was a direct Myc target and that the promoter was also coordinately down-regulated by TP53. Finally, ectopic cyclin B drove tetraploidy if the mitotic spindle checkpoint was blocked or if Myc was concurrently over-expressed [191]. These studies suggested that the loss of Mxd2 and the ensuing dysregulation of Myc predispose cells to the acquisition of tetraploidy, particularly if their cell cycle checkpoints were disrupted due to lesions such as *INK4A* and/or *TP53* inactivation [257]. Tetraploidy is likely an early, unstable and transient developmental stage that precedes the more permanent aneuploidy associated with the vast majority of epithelial cancers [258,259,260,261].

The above-mentioned negative regulatory loop between Mxd1 and the miR17-92 locus that is amplified in multiple cancers is also influenced by Mxd2 and Mnt [204,205,207,208,209,210,212,213,214]. The first intron of C13orf25, which encodes the primary unprocessed miR17-92 transcript, contains a highly conserved E-box element that binds Myc, Mxd2 and Mnt and is activated or repressed accordingly [216].

In conclusion, the role for *MXD2* as a TSG in human cancers has been controversial and perhaps obscured by the initial disregard of the importance of its haplo-insufficiency in human cancers [240,241] (Figure 2). More convincing TSG-like functions have emerged from murine models, notably those that have relied upon long-term studies and/or more sensitive techniques to detect aberrant proliferation and/or differentiation. A role for Mxd2 in suppressing non-malignant cell proliferation, such as that associated with ADPCKD deserves further investigation [242], particularly given the recently appreciated role for Myc dysregulation in the pathogenesis of this disease [243].

## 6. Mxd3

As with Mxd2 [217], Mxd3 and Mxd4 were co-identified in a yeast two-hybrid screen using Max as the bait [22]. Additionally, like Mxd1 and Mxd2, Mxd3 and Mxd4 heterodimerize with Max, bind consensus E-boxes, interact with mSin3, suppress E-box containing Myc reporters and inhibit *MYC+RAS*-mediated transformation of rat embryo fibroblasts [22].

Despite their functional similarities, *Mxd3* and *Mxd4* gene expression patterns in the murine embryo are distinct from one another as well as from those of *Mxd1, Mxd2, Myc* and *Mycn* and are mostly confined to the nervous system and skin [52]. As a general rule, *Myc, Mycn* and *Mxd2* gene expression is associated with proliferating compartments, whereas *Mxd3* and *Mxd4* are associated with more quiescent ones. However, these individual expression patterns are not absolute and vary in both tissue- and stage-specific ways [22]. For example, in the early (ca. e7.5) embryo, *Mxd3* expression is low and confined to the posterior regions of the embryonic ectoderm and the peripheral non-reactive decidua where it was co-expressed with Myc [24]. In ca. e11.5 developing limb buds, *Mxd3* is co-expressed with *Mycn* in proliferating mesenchyme adjacent to the epidermis, whereas at e17.5, undifferentiated epiphyseal growth plate chondrocytes co-express *Mxd3* and *Myc.* These differential expression patterns are maintained in species as diverse as Xenopus and are thus highly conserved throughout evolution [262].

In contrast to *MXD1* and *MXD2,* fewer studies have implicated *MXD3* in the direct pathogenesis of human cancer and/or tumor suppression. While its structural and functional similarities, coupled with mostly correlative studies, would indicate that it plays roles similar to those of its paralogs, there are some notable differences that might have been anticipated from the above-mentioned embryologic studies. For example, the Sonic Hedgehog (Shh) variant of medulloblastoma expresses high levels of Mxd3 and Mycn in the cerebellar granule neuron precursors (GNPs) or nestin-expressing progenitors from which tumor cells are thought to originate [263,264,265,266]. Indeed, Mxd3 is actually required for Shh-mediated maintenance of GNP proliferation and, in Shh’s absence, the over-expression of Mxd3 (but not other Mxd proteins) can drive GNP proliferation. This requires both Max dimerization and the interaction with mSin3 but surprisingly does not require DNA binding by Max–Mxd3 heterodimers. This suggests that Mxd3’s role is distinct from that of other Mxd members, that its impact on GNP proliferation might actually complement that of Mycn rather than antagonize it and that the Mxd3’s promotion of proliferation might be mediated by interactions outside those of the immediate Myc Network (Figure 1). For example, Max–Mxd3 heterodimers might interfere with the activity of other transcription factors in a manner that does not require direct DNA binding by Max–Mxd3 or they might be recruited to non-E-box-binding sites by such factors. Consistent with this idea are reports indicating that Mxd3 is specifically expressed during the S phase and prior to differentiation in a manner than is determined by an E2F1 binding site in its proximal promoter [22,24,266,267,268,269].

Subsequent studies have revealed Mxd3’s role to be somewhat more subtle. For example, while its short-term over-expression in DAOY medulloblastoma cells was associated with more rapid proliferation and a higher proportion of cells in the S phase, its longer-term expression actually suppressed proliferation and promoted apoptosis [266,270,271,272,273]. Comparative DNA microarray profiling between these two conditions (12 h vs. 72 h following Mxd3 induction) showed dysregulation of >2200 and >3300 genes, respectively, with considerable overlap between the groups. The uniquely-expressed early transcripts tended to encode proteins involved in immune responses, apoptosis/survival and cell cycle, whereas the uniquely-expressed late transcripts encoded proteins involved in protein folding and cell adhesion. Among the many ways to explain this switch from early to late target gene preferences is that the products of the former set of genes are responsible for remodeling the chromatin of the latter in ways that make them amenable to increased transcription.

A mechanism has been identified in murine splenic T cells by which Mxd3, acting via the Id2 protein, might regulate gene expression in a manner distinct from that of other Mxd members and potentially contribute to tumorigenesis [269]. In this setting, and in an E-box-dependent manner, the bHLH transcription factor E2A/Tcf3/E47 (E2A) was shown to drive the expression of many genes that are required for B-cell differentiation. Id2 is a HLH protein that lacks a basic domain and thus interferes with B-cell differentiation by heterodimerizing with E2A and inhibiting DNA binding [274]. Mxd3 is expressed at high levels in the most immature and highly proliferative subset of splenic T cells, binds to at least one of several E-boxes in the *Id2* gene promoter and up-regulates Id2 expression in a dose-dependent manner [269]. Mxd3’s binding to the *Id2* promoter was shown to be associated with the recruitment of p300 histone acetylase in a manner that recalled the interaction between p300 and Myc [141,269,275]. Moreover, the deliberate down-regulation of Mxd3 decreased Id2 levels, whereas the down-regulation of Mxd4 had the opposite effect. Coupled with a previous finding that Mxd3 protected against γ-radiation-induced cell death in primary thymocytes and neural progenitor cells [55], these findings suggested that Mxd3 functions in ways that are both distinct from other Mxd members and potentially pro-oncogenic. First, Mxd3 might maintain the well-known Myc-mediated differentiation block that is an important feature of tumors [276,277]. Second, the protection against DNA damage afforded by Mxd3 could dampen the apoptotic response that is triggered by the over-expression of oncogenes such as Myc itself and thereby contribute to their further dysregulation and oncogenic signaling [118,257].

Mxd3 is over-expressed in multiple cancers, including acute lymphocytic leukemia and glioblastoma (Figure 2 and Figure 3) [271] and at least 17 different tumor types show survival differences that correlate with MXD3 expression levels (Figure 3). Supporting the idea that Mxd members are functionally distinct is the finding of considerable variability among Mxd members with regard to their correlations with favorable or unfavorable survival. For example, 11 of the 14 tumor types (78.6%) expressing the highest levels of Mxd4 are associated with more prolonged survival, whereas in the case of Mxd3 this is true for only 5 of 17 tumor types (29.4%) in the case of Mxd3.

Collectively, these and the foregoing results indicate that, unlike the case for all other members of the Mxd family and despite frequent and widespread LOH in a variety of cancers (Figure 2), there is currently little additional evidence to support a TS role for Mxd3 in human cancers.

## 7. Mxd4

Like other Mxd paralogs, Mxd4 is expressed in distinct spatial and temporal patterns during differentiation [22,24,267]. Consistent with a role as a TS and inhibitor of proliferation, Mxd4 is induced within 5–8 h of initiating murine erythroleukemia (MEL) cell terminal differentiation by dimethlyl sulfoxide [278]. In this model, endogenous Myc abruptly declines to nearly undetectable levels within 1–2 h of induction, returns to pre-induction levels during the ensuing 12–18 h and then progressively declines again during the terminal differentiation phase that occurs between days 2 and 5 and is associated with proliferative quiescence, erythroid lineage commitment and the onset of globin synthesis [279,280]. Prior to differentiation, an Inr element in the Mxd4 gene proximal promoter binds and is suppressed by Miz1–Myc complexes. Upon differentiation, the decline of Myc allows Miz1 to become transcriptionally active and induce Mxd4 [278].

The above findings were extended to the in vitro differentiation of embryonal stem cells (ESCs) where both similarities and differences were noted [278,281]. Within 6 h of inducing hematopoietic differentiation by the sequential addition of bone morphogenetic protein 4, Activin A, fibroblast growth factor and vascular endothelial growth factor, a transient ~80% decline in Mxd4 transcript levels was noted versus <2-fold changes in those of Myc and other Mxds. Enforcing ectopic Mxd4 expression impaired the emergence of both primitive and committed lineages. This did not affect viability but rather led to G0/G1 cell cycle arrest in association with reductions in the cyclin-dependent kinases (Cdk) 4 and 6 and an increase in the Cdk inhibitor p27^kip1^, all of which are Myc targets [153,251,252,282,283]. A model was proposed whereby committed but actively proliferating CD41+ hematopoietic progenitors gradually suppress their expression of Myc while increasing Mxd4, which leads to cell cycle withdrawal during the latter stages of terminal differentiation [281]. The initial down-regulation of Mxd4 was proposed to allow for a transient Myc-driven proliferative burst prior to lineage specification and proliferative arrest.

Somewhat similar findings were made in senescent human fibroblasts or those rendered temporarily quiescent by serum deprivation. Both conditions increased Mxd4 transcripts levels relative to those in proliferating cells although two-fold higher levels were achieved in senescent cells [284]. Quiescent cells also significantly increased Myc transcripts and decreased Mxd4 transcripts as serum-stimulated proliferation resumed, whereas little change was seen in serum-treated senescent cells, which was consistent with their irreversible growth-arrested status. However, Miz1 bound the *MXD4* promoter and up-regulated Mxd4 expression in both senescent and serum-deprived cells [284]. In quiescent cells, serum treatment increased Myc–Max binding to several Myc target genes while also increasing the abundance of repressive Myc–Miz1 complexes on the Mxd4 promoter. In senescent cells, however, these changes did not occur. Importantly, no other Mxd members appeared to be responsive to serum stimulation in either quiescent or senescent fibroblasts thus implicating Mxd 4 as the major Myc Network factor that is responsible for mediating the permanent proliferative arrest associated with senescence and reversible arrest associated with serum deprivation.

Despite the evidence implicating Mxd4 in the induction and/or maintenance of cell quiescence, there is little published information regarding a similar role in human neoplasia. Yet it is clear from our survey of tumors from TCGA that high Mxd4 expression is associated with favorable survival in 11 cancers and with reduced survival in three (Figure 2). Mxd4 expression and its relation to long-term survival also do not appear to correlate consistently with the expression of any other members of the Extended Myc Network.

## 8. Mnt

The same successful yeast two-hybrid screen that identified Mxd3 and Mxd4 as high-affinity Max partners also isolated Mnt [51], which was soon thereafter identified independently [285]. While also being a bHLH-ZIP protein, Mnt has slightly different sequence-specific DNA binding preferences compared to other Myc Network members [51,285]. Like Mxd1–4, however, it contains an mSin3A/B-interacting SID domain that is necessary for the HDAC-mediated transcriptional repression of E-box-containing target genes and the inhibition *MYC+RAS*-mediated transformation. Subsequent work demonstrated that Mnt also forms similarly repressive complexes with Mlx [44,45,286]. Embryonic Mnt expression is broader than that of Mxd1–4 and shows tissue patterns that overlap those of Myc members, particularly in regions actively engaged in proliferation. Yet, it is also expressed in more differentiated and replicatively quiescent tissue regions that do not express other Myc Network members [51].

Like Mxd1 and Mxd4, Mnt also heterodimerizes with both Mlx and Max, thereby binding to and regulating its target genes in an E-box-dependent manner. Homodimeric Mnt also binds E-boxes, albeit relatively weakly [51,287]. It is not entirely clear how transcription is fine-tuned by each of these three different dimeric forms of Mnt. It could be a consequence of non-mutually exclusive differences in DNA binding affinities, stoichiometric variation of mSin3A/B recruitment and the degree of deacetylation of the local chromatin environment. Mnt homodimers might also be more potent transcriptional repressors by virtue of being able to recruit two mSin3 molecules versus only one each for Max–Mnt and Mlx-Mnt. For target genes whose regulation relies upon more than one E-box or on ChoREs, differential transcriptional effects might depend on the identities of the dimers bound at these sites, the proximity of these sites to one another and the factors they recruit. Any or all of these activities could be further refined by Mnt phosphorylation, which is more pronounced during quiescence and is needed for the formation of mSin3 complexes and the transcriptional repression of cell cycle entry-critical genes such as cyclin D2 [23].

Mnt participates in a negative feedback loop with its own promoter, which contains two evolutionarily conserved but non-identical E-boxes residing within 800 bp of the transcriptional start site [197,287]. Both sites bind Max but only the more 5′ and less canonical E-box does so in association with Mnt. However, Max is not necessary for Mnt-mediated repression as it still occurs in Max-negative cells, albeit more weakly [287]. A genome-wide ChIP-seq analysis in Max-negative cells has shown that Mnt binds a large number of additional genes and generally down-regulates their expression. These findings are in keeping with the observations mentioned above that Mnt dimers can be of several types. It remains unclear, however, how much Mnt-mediated transcriptional repression, both of itself and other targets in Max-negative cells, is mediated by Mnt homodimers versus Mnt–Mlx heterodimers given that both interactions were documented. Interestingly, rather than suppressing proliferation like Mxd1–4, or as it does in fibroblasts [51], Mnt was shown to support proliferation in certain Max-negative cells even when Max expression was restored [197,287]. Max contributed to Mnt’s role as a regulator of gene expression since RNAseq of Mnt over-expressing Max-negative versus Max-replete isologous cells showed nearly twice as many gene expression changes in the latter. This difference in gene target numbers may have been due in part to the fact that Mnt localizes exclusively to the nucleus in cells that express Max, whereas, in Max-negative cells, approximately half of Mnt is cytoplasmic.

Several observations support the idea that Mnt functions as a TS under some circumstances. First, in contrast to Mnt’s function in Max-negative pheochromocytoma cells mentioned above [287], *Mnt*^−/−^ MEFs grow more rapidly than their WT counterparts, are less contacted inhibited, prematurely enter the S phase, show delayed onset of senescence with continued passage, are more efficiently transformed by *MYC+RAS* and can be transformed by *RAS* alone [51,288]. These phenotypes are reminiscent of those seen in response to varying levels of Myc over-expression [289,290,291] and suggested that Mnt loss, like that of other Mxds, might rebalance the Myc Network in ways that favor Myc-driven and transformation-prone phenotypes. Indeed, *Mnt*^−/−^ MEFs over-express and/or deregulate several Myc target genes associated with cell cycle progression including cyclin E and cdk4 and possess more robust cyclinE-cdk2 activity [51].

Despite their transformation-prone tendencies, low-passage *Mnt*^−/−^ MEFs actually express less Myc than their WT counterparts even though Myc induction in response to mitogenic stimuli remains strong [51]. One possible reason for this is that there is less competition for E-box binding by Myc–Max heterodimers, leading to a more efficient induction of Myc target genes and thus a readjustment of otherwise unnecessarily high Myc levels. Nevertheless, the abnormalities in *Mnt*^−/−^ MEFs still indicate that, regardless of this presumed Myc re-balancing, the precision of its signaling remains reliant on the more nuanced regulation that can only be achieved by actual competition between Mnt–Max and Myc–Max heterodimers as well as by the regulation of genes that are Mnt targets but not Myc targets.

Conditional T-cell-specific knockout of *Mnt* prior to the divergence of CD4+ and CD8+ populations is oncogenic while concurrently leading to marked immune dysregulation [292]. Juvenile *Mnt*^−/−^ mice were found to have smaller thymuses with atrophy of the thymic medulla that is normally enriched for T cells. They also showed a differentiation block at the point of transition between the CD25+ CD44− and CD25− CD44− states (i.e., DN stages III and IV) and had peripheral lymphopenia and reduced populations of CD4+CD8+, CD4+ and CD8+ thymocytes associated with higher levels of apoptosis. In keeping with the general theme described for *Mnt*^−/−^ MEFs [51], *Mnt*^−/−^ thymocytes up-regulated cell cycle-related Myc target genes, including those encoding cdk4, cyclin D2 and cyclin B1 [191,292]. These mice also developed hepatosplenomegaly and lymphadenopahthy at an early age. Initially, this was associated with the uncontrolled infiltration of non-transformed B cells and macrophages driven by the secretion of cytokines such as IFN-γ, TNF-α and IL-2 elaborated by *Mnt*^−/−^ Th1 cells. By 12–22 months of age, however, most of these mice succumbed to T-cell lymphomas, although their onset was slower relative to the lymphomas seen in *Myc* transgenic animals that arose as early as 2 months of age [293]. This indicated that *Mnt* loss and *Myc* over-expression were not interchangeable, probably because the levels of Myc in the transgenic setting were much higher. As a result, they bound and regulated different sets of target genes, particularly those with low-affinity E-boxes in the case of Myc transgenics. The aggressive immune dysregulation and expansion of B cells and macrophages seen in *Mnt*^−/−^ T cells have also not been previously reported in association with Myc-driven lymphomas.

A different role for Mnt was seen when it was conditionally inactivated in T cells that also constitutively over-expressed a stabilized and particularly oncogenic form of Myc (Myc^T58A^) [294]. These *Mnt*^−/−^ mice showed a 1.6-fold longer mean survival than was seen in Myc^T58A^ *Mnt^+/+^* mice. Thus, rather than serving as a TS, Mnt functioned as a pro-oncogenic tumor facilitator in cooperation with Myc. *Mnt*^−/−^ T cells were also more susceptible to apoptosis. This suggested that cells with Myc deregulation walk a fine line between transformation and apoptosis with the highest levels of Myc expression favoring the latter fate. By blocking apoptosis, Mnt could cooperate with Myc to allow more cells to be directed down the transformation pathway.

Some support for these notions comes from studies in mice that developed distinct types of hematopoietic malignancies due to the over-expression of different levels of Myc driven by the vav-promoter [295]. Those with two copies of the vav-Myc transgene (“Mychi”) developed pure T-cell lymphomas, whereas those with only a single copy of the transgene (“Myclo”) developed both lymphoid and myeloid neoplasms [295,296]. Hypothesizing that Mnt would function as a TS, it was predicted that *Mnt^+/−^* Myclo mice would have enhanced Myc function, generate a higher proportion of the lymphoid neoplasms otherwise associated with Mychi mice and would succumb earlier [295]. Instead, virtually all tumors generated in *Mnt^+/−^* Myclo mice were myeloid and associated with longer rather than shorter survival. Similar studies performed in *Mnt^+/−^* Mychi mice also showed a higher than expected fraction of myeloid neoplasms and extended survival. Thus, under these conditions, Mnt facilitated Myc-driven tumorigenesis. An extension of these studies to Eμ-Myc transgenic mice revealed that *Mnt* haploinsufficiency was not associated with any change in B-cell lymphoma phenotypes but did extend survival by 1.7-fold. Additionally, no differences were observed in the apoptotic susceptibility of the pre-leukemic B-cell populations of Eμ-*Myc/Mnt^+/+^* and Eμ-*Myc/Mnt^+/−^* mice to IL7 deprivation nor did tumors show any differences in the fractions of cells undergoing S-phase transition, apoptosis or senescence. Finally, *Mnt* heterogeneity prolonged rather than reduced survival due to spontaneous tumorigenesis in *Tp53^+/−^* mice.

In contrast to the above observations in hematopoietic tissues, female mice bearing a mammary epithelium-specific knockout of *Mnt* were prone to the development of breast adenocarcinoma [53,297]. Moreover, and unlike the more delayed appearance of lymphomas that arise in *Mnt*^−/−^ mice described above [295], the breast cancers developing in *Mnt*^−/−^ and *Myc* over-expressing mice showed similar latency periods [53,297,298]. Like *Mnt*^−/−^ MEFs, *Mnt*^−/−^ breast epithelium expressed higher levels of cyclin E and cdk4 that increased further following transformation. Together, these findings implicated shared biochemical features of *Mnt*^−/−^ cells that correlate with deregulated growth and transformation. This is further in keeping with the idea that, because Myc–Max and Mnt–Max complexes seem to comprise the most common heterodimeric species in most cells, Mnt’s loss might wield a disproportionate impact on growth.

*MNT*’s function as a TSG in humans may be quite common given the frequency of *MNT* LOH in human cancers although the impact on survival is variable (Figure 2 and Figure 3). For example, a significant fraction of breast cancers harbor *MNT* gene deletion within the chromosome 17p13.3 locus that do not include the nearby *TP53* gene located on 17p13.1 [299,300]. Where it has been examined, most such *MNT* deletions are not associated with mutation of the retained allele, which again raises questions about alternate means of *MNT* gene inactivation or the role of *MNT* haplo-insufficiency such as that discussed above in mouse models [295,301]. One means by which the former could be achieved is via the up-regulation of miR-210, which occurs under hypoxic conditions and is over-expressed in numerous cancers, particularly those associated with metastases or unfavorable survival [302]. Mnt mRNA contains multiple miR-210 binding sites in its 3′-untranslated region and knockdown of the transcript phenocopies miR-210 over-expression [302]. Such alternative means of inactivating other Mxd members, including epigenetic ones, have not been systemically investigated in human cancers.

Sezary syndrome is a rare but aggressive cutaneous T-cell lymphoma of CD4+ memory cells that is molecularly heterogeneous [303,304,305,306,307]. In a study of 20 patient that employed high-resolution comparative genomic hybridization, *MYC* copy number gains were detected in 75% of leukemic cell samples, *MXD2* loss in 60% and *MNT* loss in 55% [307]. These commonly occurred together, with *MYC* amplification and *MXD2* loss being seen in seven samples, MYC amplification and *MNT* loss being seen in nine samples and *MYC* amplification and loss of both *MXD2* and *MNT* loss being documented in six samples. Interestingly, the solitary loss of *MXD2* or *MNT* was found only once. A subsequent European and multi-institutional study of Sezary syndrome samples utilized a targeted, quantitative PCR-based approach that examined gains and losses in a prescribed set of genes [308]. The study found a 40% incidence of *MYC* gain and a 66% incidence of *MNT* LOH in 58 samples. Although Mnt expression was not quantified, none of the 58 samples showed evidence of Myc over-expression, regardless of gene copy number. Taken together, these findings further contribute to the idea that the rebalancing of the Myc Network can involve contributions from more than one member (Figure 2).

At first glance, the findings in Sezary syndrome seemingly contradict the previously discussed murine studies showing that Mnt is pro-oncogenic, in collaboration with Myc^T58A^ transgene over-expression [295,296]. There are potentially several non-mutually exclusive explanations for this. For example, the murine studies were conducted in the background of Myc^T58A^ dysregulation. In this context, any TS activity of Mnt might have been obscured or over-ridden, either as a direct result of Myc dysregulation or as a more indirect consequence of lineage selection within the hematopoietic cell environment that promoted myeloid rather than lymphoid tumors. Myc^T58A^ over-expression also likely altered a subset “non-physiologic” Myc targets. Therefore, the seemingly contradictory murine and human findings are not necessarily incompatible. Rather, they are entirely consistent with the findings that Mnt (as well as other Extended Myc Network members) can serve dual functions as oncogenes or TSGs, association with both favorable and unfavorable outcomes, depending on the context (Figure 2 and Figure 3).

A high-resolution single-nucleotide polymorphism array analysis of more than 350 previously untreated chronic lymphocytic leukemias (CLLs) identified a mean of 1.8 CNVs per sample, with deletions being nearly 4-fold more frequent [309]. In total, 8% of cases involved focal *MNT* LOH at 17.13.3 although the vast majority of losses were considerably larger, encompassing as much as the entire chromosomal arm. An additional 4% of cases showed LOH of the *MGA* gene locus on 15q15.1. Finally, 5% of cases showed *MYC* amplification although, reportedly, none showed actual evidence of Myc over-expression despite this often being associated with shorter survivals in a variety of B-cell neoplasms, including CLL (Figure 3) [77,310,311,312,313]. This study again hinted at the possibility that *MNT* and *MGA* haplo-insufficiency might be sufficient to contribute to and alter clinical courses in their role as actual TSGs.

Recent evidence has pointed to a role for Mnt, and perhaps other members of the Extended Myc Network, in the TGF-β-induced epithelial-to-mesenchymal transition (EMT) of normal murine and human breast epithelium. Mnt was highly induced during this process and its knockdown increased cellular proliferation and blocked EMT [314]. Mnt was also expressed at higher levels in human triple negative breast cancer (TNBC) cell lines, which typically show more marked EMT and a predisposition to invasive and metastatic behaviors [315,316,317]. The above study also used the expression of a group of established EMT markers to classify breast cancers from TCGA into those with “EMT high” or “EMT low” features [314]. As had been seen previously in TNBC cell lines, EMT high tumors also expressed the highest levels of Mnt, whereas the reverse was true for EMT low tumors. Although no evidence was presented to show whether Mnt expression correlated with survival, our analyses of the same TCGA dataset indicates that it does not (Figure 3) [314]. Collectively, these studies suggest that the higher expression of Mnt associated with EMT in breast cancer actually tempers proliferation and thus serves more as a TS. Moreover, the simple model that *MNT* amplification might be pro-oncogenic is belied by the finding that gene copy number loss rather than gain is much more frequent in breast and many other cancers (Figure 2). Nevertheless, it is still important to consider the potentially pro-oncogenic consequences of Mnt over-expression, which would tend to favor EMT both in breast and other cancers where it often contributes to such unfavorable behaviors as metastatic dissemination and chemotherapy resistance [318,319,320].

A recent comparison of RNAseq results from control and Mnt-knockdown normal mammary cells undergoing TGF-β-induced EMT showed that Mnt negatively regulated 1180 genes and positively regulated 678 genes [314]. The former set was enriched for transcripts encoding epithelial markers, whereas the latter was enriched for transcripts encoding mesenchymal markers. ChIP-seq confirmed the interaction of Mnt with numerous genes that were negatively regulated by Mnt and co-IP experiments showed that Mnt interacted with mSin3A and HDAC1, while promoting the deacetylation of histone H3K27.

The above experimental studies suggest that Mnt’s dual roles as either a TS or tumor facilitator are fluid and dependent-in at least some cases-upon the degree of Myc over-expression, the tissue-specific target gene repertoire and the sensitivity of the target tissue to genes that promote apoptosis and transformation. On the one hand, Mnt may impede oncogenic growth by competing for and suppressing critical Myc target genes. Alternatively, as Myc levels rise and pro-apoptotic genes are summoned, Mnt may facilitate tumorigenesis by blocking apoptotic pathways or altering the balance of competing factors leading to EMT [53,118,288,289,290,294,314,321,322]. The ensuing higher levels of Myc expression may have secondary consequences beyond its direct ones on gene expression, including the induction of ROS and chromosomal instability that can further contribute to tumor pathogenesis and/or evolution as well as tumor cell apoptosis or senescence [256,257,323]. In some circumstances, Mnt’s suppression of apoptosis may facilitate Myc-mediated tumorigenesis as happens in collaboration with the over-expression of Bcl-XL or Bcl-2 or the knockout of *TP53* [324].

## 9. Mga

Mga is yet another Max partner that was identified by yeast two-hybrid screening [22,51,52,53,54]. It is an extremely large protein (3065 amino acids) that, in addition to containing a canonical C-terminal bHLH-Zip domain most closely related to that of Myc, also contains an N-terminal ~185 residue ‘T domain’ that is conserved among the evolutionarily related T-box transcription factors that include Brachyury and the Tbx family members [325]. Between these two regions is a ~300 residue ‘DUF4801’ domain that until recently has been largely uncharacterized [326]. Like the bHLH-ZIP domain, the T domain, which most resembles that of Tbx6, functions in both dimerization and DNA binding [52,327,328,329]. Thus, Mga’s design immediately suggested that it might possess unusually versatile DNA binding and transcriptional functions by participating in the regulation of both E-box- and T-box containing genes.

E-box binding by Mga requires Max and its basic domain and probably requires Mga’s basic domain as well although this has not been tested directly (Figure 5A) [52]. T-box-mediated activation also requires both Mga and Max but not Max’s basic domain (Figure 5B). This suggests that Max’s role in T-box-containing gene activation is to facilitate the recruitment of other T-box-specific transcription-critical factors in a manner analogous to that of Myc [16]. The up-regulation of a reporter gene containing both E-boxes and T-boxes has been shown to be additive when Mga and Max are co-expressed but only if Max’s basic domain is present to allow E-box binding (Figure 5C). Finally, in Max’s absence, Mga actually repressed transcription from a T-box-containing reporter (Figure 5D) [52]. These studies indicate that Mga’s function is complex. Firstly, it may behave like Myc with regard to its interaction with Max and its transactivation potential. Secondly, as the only member of the Brachyury/Tbx family with a bHLH-ZIP domain, Mga can bind to and regulate two distinct classes of genes [330]. Finally, whether Mga serves as a transcriptional activator or repressor appears to depend strongly on Max levels.

Despite its Myc-like function as a transcriptional activator, Mga was found to suppress *MYC+RAS*-mediated fibroblast transformation [52]. This required an intact bHLH-ZIP domain and could be reversed by sufficient amounts of Max. This suggested that Mga functionally blocks Myc by competing for Max as well as for E-box and/or ChoRE occupancy. In keeping with the fact that no other member of the Brachyury/Tbx family contains bHLH-ZIP domains [330], Brachyury exerted no influence on *MYC+RAS*-mediated transformation [52].

Tbx proteins are intimately involved in multiple stages of embryonic differentiation and, like the Myc Network, regulate their target genes in distinct but overlapping stage- and tissue-specific ways [329]. The murine embryonal expression pattern of Mga closely mimics that of Brachyury and Tbx’s 2–5, thereby suggesting that it might cooperate with these proteins in the control of mesoderm induction, while being particularly well-positioned to integrate and coordinate these processes with the Myc Network [52]. Investigations of Mga’s role during murine embryogenesis have shown Mga to be required for survival of the epiblast of the pre-implantation embryo. *Mga* hypomorphs survive longer but still died early in gestation with reduced development of pluripotent cells in embryonic regions [331,332].

CNVs of the *MGA* locus (15q15.1) occur in many human cancers, with virtually all cases involving single copy loss (Figure 2). For example, the previously cited SNP array survey of CLL samples found a 4% incidence of 15q.15.1 heterozygous deletion with only a single sample showing inactivation of the retained allele [309]. Moreover, as seen recurrently for other members of the Extended Myc Network, the correlation between Mga expression and survival, when it does occur, is highly tumor specific and may correlate with either favorable or unfavorable survival (Figure 3).

Another study of CLL that addressed the potential role of MGA haploinsufficiency involved 55 previously treated individuals with high-risk disease and 33 with Richter’s syndrome (a particularly aggressive form of CLL associated with rapid conversion into DLBCL [333]. Somatically-acquired and inactivating *MGA* mutations were identified in 5.4% of cases and copy number loss was identified in an additional 4%, with none of these being correlated with survival. A larger study in 288 CLL patient, 65% of whom had developed relapsed or refractory disease, found a 4% incidence of somatically-acquired MGA mutations, with a small but significant tendency among this group to have a worse overall response to treatment [334].

In two studies involving 105 and 71 patients with untreated, EBV-associated NK-cell lymphoma, an aggressive neoplasm involving CD56+ cytoCD3+ lymphocytes, *MGA* mutations were identified at frequencies of 8.8% and 4%, respectively [155,335,336]. However, neither study determined the consequences of these mutations or whether they correlated with survival.

In addition to the above and other reports in CLL, *MGA* inactivation, mostly involving terminating mutations or gene deletion (and nearly always hemizygous), have been described in as many as 10–20% of a variety of cancers including retinoblastomas, sporadic colorectal cancers, gastroenteropancreatic neuroendocrine tumors, gastrointestinal stromal cell tumors, lung squamous cell and adenocarcinomas, head and neck cancers and acute myelogenous leukemias (AML) [29,30,104,117,333,337,338,339,340,341,342,343,344,345,346]. Silencing Mga in an AML cell line increased both its in vitro colony formation and its growth rate [344]. Supporting the idea that Mga–Max heterodimers may compete with Myc–Max heterodimers for E-box occupancy [52], knockdown of Mga increased the expression of a Myc reporter vector [344]. This work also confirmed that AML blasts expressing higher levels of Mga are associated with favorable survival (Figure 3) [344]. A reasonable expectation of this result is that patients with AML whose leukemic blasts express lower levels of Myc would also have more favorable survival but this was not observed using data from TCGA (Figure 3). In contrast, a more recent study of 265 previously untreated AML patients did show a near doubling of survival for those whose blasts expressed the lowest levels of Myc [347]. These differences may have been attributable to different patient populations and/or to the fact that the latter group measured actual Myc protein levels rather than transcript levels, which as discussed above, do not always correlate. Features of the ‘low Myc’ cohort that were proposed as contributing to longer survival included lower numbers of both peripheral and bone marrow blasts and lower percentages of Flt3-internal tandem duplications [347,348].

An interesting racial disparity has been observed in a large comparative study of recurrent mutations associated with DLBCL in Black African versus Caucasian individuals. This revealed deleterious *MGA* mutations at a 3.7-fold higher incidence in the former cohort (19.7% vs. 5.33%, *p* < 0.001) [349]. *Mga* also registered as one of the top hits in a genome-wide CRISPR screen for TSGs in DLBCL [350].

The above studies clearly imply that Mga functions in vivo and in numerous clinical settings as a TS, albeit with tumor-specific differences on survival (Figure 2 and Figure 3). This has been investigated in studies that attempted to ascertain the mechanism underlying this behavior. To this end, the composition of endogenous Mga protein complexes from human HEK293T and lung adenocarcinoma cells have recently been analyzed by immunoprecipitation and mass spectrometry. In addition to Mga itself and Max, over 80 additional associated proteins were identified [29]. Many of these play roles in transcriptional repression including the dimeric pair of E2F6 and TFDP1; components of the Polycomb Repressive Complex 1 (PRC1) complex such as RNF2, RING, L3MBTL2 and PCGF6; the histone deacetylases HDAC1 and HDAC2; and the heterochromatin-associated chromobox protein CBX3, which recognizes the H3K27me3 epigenetic markers left by PRC2 [351,352,353,354,355]. ChIP-seq studies performed in lung adenocarcinoma cells and surveys of the ENCODE database also showed the genomic binding sites for Mga to be enriched for Max and virtually identical to Myc binding sites. Other Mga binding sites were enriched for E2F6. An interesting finding was that ChIP with anti-Mga antibodies was not reported to show evidence for binding to T-box elements, thus indicating that transcriptional regulation by Mga, at least in the cell lines examined, was primarily centered around Myc targets. These results indicated that, in several normal and neoplastic cell types, DNA-binding complexes comprised of Mga, Max and E2F6 interact with members of the PRC1 complex, compete with Myc–Max complexes for binding to Myc-activated targets and repress their expression.

The above study also examined the gene expression profiles of Mga over-expressing A549 lung adenocarcinoma cells and identified 625 repressed genes and 658 activated genes [29]. Gene set enrichment analysis of the former indicated that they tended to contain direct Myc targets, whereas Mga-activated genes tended to be comprised of a distinct group that was five-fold less likely to contain Myc targets. Nevertheless, both gene groups tended to be associated with the same types of Myc-like cellular activities such as cell cycle control and glycolysis. Finally, Mga over-expression and siRNA-mediated Myc knockdown had indistinguishable effects with regard to their ability to inhibit cell proliferation, thus further attesting to the idea that the TS-like consequences of Mga over-expression are equivalent to those obtained following Myc depletion [29]. The fact that Mga–Max binding to direct Myc targets could be more readily identified in A549 cells when Mga was over-expressed suggested that these targets contained low-affinity Mga–Max sites although this was not tested directly.

A more recent study of Mga function in cancer that expanded upon the above results indicated that CRISPR-mediated Mga inactivation accelerated tumor growth and shortened survival in both the KrasLSL-G12D and KrasLSL-G12D+*Tp53*^−/−^ models of lung cancer [29,30]. While this indicated that Mga loss facilitated the growth of tumors in highly susceptible hosts, it was not determined whether Mga inactivation in otherwise normal mice impacted spontaneous tumor growth as happens with more traditional TSs such as TP53 or Mlx [76,356,357,358].

Whole transcriptomic comparison of KrasLSL-G12D tumors with intact or inactivated Mga indicated that the latter up-regulated many genes previously identified as Mga-PRC1.6-repressed targets in ES and germ line cells [359] cells, including numerous Myc target genes. Mga-inactivated tumors also down-regulated genes involved in the anti-tumor response such as those encoding NK cell markers and interferon signaling pathway components. Similar analyses performed on cell lines established from KrasLSL-G12D+*Tp53*^−/−^ tumors supported the previous findings with regard to Myc target genes and Mga-PRC1.6-repressed genes while also showing a substantial up-regulation of genes related to TGF-β signaling and EMT. These latter findings implicated Mga in repressing invasion and maintaining epithelial identity while also demonstrating that the identities of Mga-regulated genes can be altered by the tumor’s *TP53* status.

Mga-interacting proteins identified by mass spectrometry of immunoprecipitates from 293FT cells included some of those previously described [29], including the ncPRC1.6 complex-associated proteins L3MBTL2 and RING2 as well as HDACS1 and 2 and components of the chromatin-modifying MLL-WRDA complex such as WDR5, RBBP5, ASH2L, and DPY30. Cells lacking Mga also expressed lower levels of Max, E2F6 and L3MBTL2, which have been previously identified as being stabilized by the ncPRC1.6 complex [352,353,360]. The functionally uncharacterized DUF4801 domain of Mga was also shown to serve as a scaffold for the ncPRC1.6 complex as well as to suppress tumor cell growth.

A recently performed ChIP-seq survey identified the genomic sites occupied for Myc, Max, Mga, L3MBTL2, E2F6 and phosphorylated RNA pol II in Mga-replete and Mga-knockout KrasLSL-G12D+*Tp53*^−/−^ tumor cells [30]. Components of the Mga-PRC1.6 complex (i.e., Mga, Max, L3MBTL2 and E2F6) bound several thousand promoters in Mga-replate cells with the vast majority of Max-, E2F6- and L3MBTL2-binding sites also being bound by Mga. In cells lacking Mga, Myc–Max binding increased for a subset of binding sites but otherwise remained largely unaltered. This stability of the Myc binding landscape could have been due to sites no longer occupied by Mga now being replaced by Mxd–Max or Mnt–Max rather than Myc–Max, to the newly exposed sites being unfavorable to Myc–Max binding in general and/or to factors that directly blocked Myc binding. Further consistent with the relative lack of influence of Mga depletion on Myc binding was the finding that, rather than increasing, the association of RNA pol II with previous Myc-bound genes actually decreased somewhat. The depletion of Myc from either Mga-replete or Mga-inactivated KrasLSL-G12D+*Tp53*^−/−^ or A549 lung adenocarcinoma cells equally suppressed proliferation, whereas the depletion of Mga, L3mbtl2 or Pcgf6 had no impact.

These results were extended to normal colonic epithelium and colorectal cancer, in which *MGA* is frequently mutated or deleted (Figure 2) [30]. In the former case, CRISPR-mediated *MGA* inactivation in colon organoids was associated with reduced levels of L3mbtl2 and an increased rate of 3D growth in vitro. In addition to the up-regulation of several EMT-associated genes, and an array of E2F-regulated genes involved in cell cycle regulation and DNA replication, four of the five genes that were up-regulated in response to *MGA* inactivation in lung cancer (STAG3, PODXL2, NHLRC1, and ZCWPW1) were also up-regulated in *MGA*-knockout colonic organoids. Additionally, in keeping with the findings in KrasLSL-G12D tumors, down-regulated gene sets included those involved in interferon signaling and inflammation.

The above findings raise several questions regarding the role(s) played by Mga–Max and Myc–Max in regulating Myc targets and Myc-driven functions. For example, to what degree do different tissues and transformation impact the target gene repertoires of these heterodimers? How do the identities of Myc targets recognized by Mga and the extent of their suppression differ from what is achieved by Mxd1–4 or Mnt? How is the regulation of Myc targets with multiple E-boxes impacted by the binding of only a single type of heterodimer versus the binding of other heterodimers? What determines whether Mga–Max heterodimers will activate or repress target genes, despite binding to E-boxes in both? Is it a consequence of subtle differences in the E-box sequences, the presence of other factors bound nearby or at more distant sites or tissue-specific factors that can dictate the identities and stoichiometries of the numerous proteins that associate Mga–Max heterodimers? Finally, depending upon the circumstances, the T-box of Mga seems capable of either strongly activating transcription or not activating transcription at all [29,52]. Other than the presence of T-boxes, what determines the relative strength and dominance of these effects and how does cross-talk with other Myc Network members bound at nearby sites fine tune them?

## 10. ChREBP

ChREBP was originally identified in rat liver by its ability to bind directly to and up-regulate the liver-type pyruvate kinase gene (*Pklr*) [58]. This occurred via ChREBP’s binding to a ChoRE in the gene’s promoter (Figure 1) and was facilitated by glucose and/or glycolytic intermediates such as glucose-6-phosphate [35,58]. Under these conditions, transcriptional activation relied upon the insulin-independent binding of glucose to cytoplasmically resident ChREBP which was then dephosphorylated by protein phosphatase 2A and translocated to the nucleus where it bound to and activated its target gene repertoire [361]. ChREBP was proposed to be important for optimizing glucose uptake and glycolysis, for converting carbohydrates into fats and for long-term energy storage. A number of other ChREBP-responsive ChoRE-containing genes have been identified such as acetyl CoA carboxylase, fatty acid synthase and other lipogenic genes [73,84,362,363,364,365]. Just as Myc’s transcriptionally active state involves its heterodimeric association with Max, ChREBP requires a similar association with Mlx [71,73,74]. In contrast to Myc, however, the target gene repertoire for ChREBP–Mlx heterodimers is more restricted, which is likely to be at least partially related to the more complex nature of the ChoRE (Figure 1). Rather than the originally proposed consensus sequence (CACGTG)N_5_(CACGTG), a more ambiguous one (CAYGNG)N_5_(CNCRTG) has been proposed more recently [71]. Moreover, ChREBP–Mlx heterodimers can also bind to single E-boxes in a glucose-independent manner [72,73,74]. The fact that Myc can also bind the double E-boxes of the ChoRE suggests that shared gene targets may allow for transcriptional promiscuity and additional levels of cross-talk among members of the Extended Myc Network [76].

Several studies have identified domains within ChREBP’s N-terminal “glucose-sensing module” (GSM) that are responsible for glucose-binding and transactivation. The GSM contains five so-called Mondo Conserved Regions (MCRs) that are required for the protein’s binding of and response to glucose [366]. When glucose concentrations are low, MCRs I–IV, and MCR IV in particular, interact with and suppress the MCR V-containing transactivation domain (also known as the GRACE domain). Glucose relieves this inhibition in a manner that appears to be more reliant on MCRs I–III. The glucose responsiveness of ChREBP however also requires the interaction between MCR3 and members of the 14-3-3 protein family that include its β, γ, ζ and θ isoforms [367,368]. The 14-3-3 proteins are an important class of signaling proteins that bind many important proteins such as TP53, various phosphatases and Raf and MAP kinases with roles in critical functions such as growth, proliferation and survival [369,370]. The 14-3-3 proteins appear to be involved in reversing or preventing the suppression of MCR IV on the GRACE domain. However, the interaction between MCR IV and 14-3-3 requires neither glucose nor the recognition of phosphoryated Ser and Thr substrates to which 14-3-3 proteins typically bind in other targets. It has been suggested that 14-3-3 plays a permissive role, possibly by allowing ChREBP to assume the conformation needed to relive its transcriptional repression of the GRACE domain in response to glucose [366]. This raises the possibility that concurrent 14-3-3 recognition of phospho-Ser/Thr sites on active signaling pathway intermediates might coordinate the glycolytic and fatty acid synthetic responses mediated through ChREBP.

The interdependence of Myc, ChREBP and the Mlx Network in general for both normal and cancer cell proliferation has been demonstrated in studies that employed the previously-discussed mouse model of hereditary tyrosinemia [41,76,78,371]. Unlike *Myc*^−/−^ hepatocytes, *Chrebp*^−/−^ hepatocytes were defective in repopulating the liver. However, the combined knockout of both *Myc* and *ChreBP* was additive and the knockout of both *Myc* and *Mlx,* which functionally inactivated both pathways (Figure 1), was the most defective of all [41,76,78]. The importance of these deficits on neoplastic growth was further demonstrated in *Myc*KO, *Chrebp*KO and *Myc*KOx*Chrebp*KO livers in which HBs were generated by mutant forms of β-catenin and YAP [41,78,168,213,239]. Transcriptional profiling of KO livers and tumors from these groups showed that the proliferative defects correlated with the suppression of numerous genes related to protein translation and metabolism. In the first case, these included virtually all of the ~80 ribosomal subunit genes, many genes controlling translational initiation, elongation and termination and additional genes whose products are involved in tRNA and rRNA biosynthesis and processing [41,76,78]. Genes involved in metabolism included those regulating mitochondrial structure and function along with mitochondrial ribosomal proteins and components of the TCA cycle and electron transport chain. These findings indicated that maximizing normal and neoplastic hepatocyte growth was variably dependent on cross-talk between the Myc and Mlx Networks. Importantly, while neither of these was necessary for tumor initiation, each one was important for maximizing the expression of the above gene sets in order to achieve the levels of expression commensurate with proliferative demands. This likely reflected the fact that protein translation and proliferation are among the most energy-dependent processes of normal and neoplastic cells and must be coordinated accordingly [91,372]. A similar role for ChREBP has been described in pancreatic β cells where ChREBP is required for their proliferative expansion in response to insulin signaling [373].

While there is little published information as to whether ChREBP expression impacts the behavior of human cancers as it does in the case of murine HBs [41], our survey points to clear associations in at least 18 different tumor types (Figure 2 and Figure 3). For example, elevated ChREBP expression is associated with favorable survival in epithelial tumors as diverse as bladder, liver and pancreatic cancer, whereas the reverse is true for LGGs, AML and KIRC. One interesting finding, although of currently unknown significance, is that DLBCL is the only cancer for which the expression of only one member of the Extended Myc Network (ChREBP) is predictive of survival. These findings again emphasize that, as is true for the Myc Network, different cancers show distinct patterns of Mlx Network member expression and survival correlations that likely reflect their differential co-dependencies.

## 11. MondoA

Originally identified by yeast two-hybrid screening as a bHLH-ZIP binding partner for Mlx, MondoA was quickly determined to be a paralog of ChREBP, albeit with a wider tissue distribution pattern including particularly high expression in skeletal muscle, where it is required for normal development [31,56,88,89,374]. Like ChREBP, MondoA is regulated by glucose and G6P in a manner that utilizes a similar N-terminal GSM module that is also dependent upon 14-3-3 protein interactions [35,375]. In addition to association with cytoplasmic lipid droplets, MondoA associates with the outer mitochondrial membrane, suggesting that it is well-positioned to respond to both glycolytic and TCA cycle substrates and their fluxes [40,62,63]. Its rapid nuclear translocation in response to glucose or G6P requires prior heterodimerization with Mlx followed by glucose-dependent target gene promoter binding of the heterodimers and their recruitment of a histone H3 acetyltransferase that restructures chromatin in preparation for gene activation [37,62,63]. MondoA–Mlx binding then directly induces the transcription of at least three glycolytic genes, namely, those for hexokinase II, 6-phosphofructo-2-kinase/fructose-2,6-bisphosphatase and lactate dehydrogenase A, although in a manner that recognizes consensus E-boxes rather than ChoREs [40]. Given that *the PKLR* gene is also a target for ChREBP and Myc, and likely for MondoA as well [68,76,77,117,302,313,346], these findings suggest that significant control over glycolysis is supervised by both arms of the Extended Myc Network in ways that match proliferation with appropriate levels of glucose oxidation, ATP production, mitochondrial function and anabolic substrate supply.

In addition to the above glycolytic targets, two paralogous genes, namely thioredoxin-interacting protein (*TXNIP*) and arrestin domain-containing protein 4 (*ARRDC4*), are among the most highly MondoA-responsive in HA1E renal epithelial cells [62,63]. *TXNIP* induction is direct and due to the binding of MondoA–Mlx heterodimers to two consensus ChoREs in the gene’s promoter [21,62,63]. As a thiol-oxidoreductase, TXNIP protein regulates redox balance, primarily by reducing oxidized cysteine and cleaving disulfide bonds [376]. However, it also counters the pro-glycolytic effects of MondoA–Mlx by down-regulating glucose uptake in skeletal muscle and other cell types [62]. Thus, the induction of glycolytic genes by MondoA in response to intracellular glucose and G6P is balanced by MondoA’s indirect inhibition of glucose uptake mediated by TXNIP and presumably ARRDC4 as well. A potential connection between TXNIP’s role as glycolytic regulator and that of an oxido-reductase has been demonstrated by studies showing TXNIP to be inducible by adenosine-containing molecules, including adenosine itself, ATP and NADH in ways that cooperate with glucose [59,67]. A more extensive role for TXNIP (and indirectly for MondoA and ChREBP) as a TS in several cancer types has recently emerged although it appears to involve more than simply its negative regulation of glucose uptake [68,377]. For example, TXNIP stabilizes the cyclin-dependent kinase inhibitor p27^KIP1^ [378]. While a similar TS role for ARRDC4 is lacking, recent work indicates that its paralogs ARRDC1 and ARRDC3 may fulfill a similar function [379,380,381,382].

In our TCGA survey, MondoA expression correlated with extended survival in three different cancer types and with shortened survival in nine (Figure 3). However, there was little consistency with regard to gene CNV across a broad range of tumors (Figure 2). In contrast, ChREBP expression correlated with favorable survival in nine cancer types and with unfavorable survival in an additional nine, with copy number gains rather than losses tending to be considerably more common. These findings further underscore the notion that MondoA and ChREBP, while perhaps being redundant, are not entirely interchangeable. Rather they perform distinct functions at specific times during development and in tissue-specific ways that are likely influenced by various glycolytic and other metabolites, the level and the source of ATP (glycolysis versus oxidative phosphorylation [Oxphos]) and the redox environment [36,45,46].

In a small, microarray-based study of 25 primary pediatric acute pre-B-cell leukemias (ALL), extremely high levels of MondoA expression were found in leukemic blasts relative to fetal pre-B cells and numerous normal tissues, including peripheral blood and bone marrow (*p* = 1.6 × 10^−21^) [96]. Additionally, noted was a strong correlation between MondoA and Mlx expression. MondoA over-expression was highly specific for pre-B ALL and was not observed in more than 200 cases of AML. Profiling of Nalm6 ALL cells line after stable genetic knockdown of MondoA showed the dysregulation of nearly 200 genes with selective roles in metabolism, differentiation and survival. A number of the down-regulated genes corresponded to those previously identified in primary ALL blasts as being associated with better therapeutic outcomes and survival [383]. Even more strikingly, cells with a genetic knockdown of MondoA demonstrated reduced clonogenicity; lower glucose utilization; a 5- and 19-fold up-regulation of the B-cell differentiation markers CD22 and CD24, respectively, and a higher levels of spontaneous apoptosis. These studies supported the idea that MondoA maintains a differentiation block in ALL blasts. Consistent with this finding, a subsequent study showed leukemic blasts with the highest levels of MondoA were associated with shorter survival [95].

In contrast to the foregoing study that ascribed a pro-oncogenic function to MondoA, evidence for TS-like activity was provided by studies which examined glycolytic rates of melanomas bearing oncogenic B-Raf^V600E^ mutations prior to and following treatment with the B-Raf inhibitors vemurafinib and dabrafenib [384]. This study observed that melanomas have high glycolytic rates, which is consistent with their known avid uptake of ^18^F-deoxyglucose [385,386,387]. Treatment with venurafinib potently suppressed glucose uptake in a manner that correlated with inhibition of the B-Raf→MEK→ERK pathway. Transcriptional profiling revealed that vemurafinib inhibited the induction of the glucose transporter genes *GLUT1 (SLC2A1)* and *GLUT3 (SLC2A3)* and hexokinase 2 (*HK2*), whose encoded enzyme catalyzes the first step of glycolysis. Significant down-regulation of these three genes was observed in multiple primary B-Raf^V600E^ melanomas from patients treated with B-Raf inhibitors and normalization of their expression in drug-resistant tumors. B-Raf inhibition also induced *MONDOA, TXNIP* and *ARRDC4*. These studies indicated that B-Raf^V600E^ normally suppresses *MONDOA* expression, which would be expected to benefit the tumor given that TXNIP inhibits glucose uptake [62,63,94].

The circumstances under which MondoA facilitates or suppresses cell growth may be determined by the fine balance between glycolysis, Oxphos and the rapidity of cell growth, which may be fleeting in nature and spatially and temporally different, even within the same tumor [388]. In turn the glycolytic rate may be modulated by competition between glucose uptake and oxidation, which are regulated in opposite ways by MondoA and ChREBP as discussed above. High glycolytic rates are needed to supply anabolic precursors such as amino acids and ribose sugars while generating energy and maintaining the proper redox state to maintain these reactions [389,390]. Oxphos must also be maintained (although typically at a reduced rate) to furnish ATP more efficiently than is possible by glycolysis and to supply other anabolic intermediates. One way to achieve this is via glutaminolysis which anaplerotically provides α-ketoglutarate (α-KG) [25,26,391]. This TCA cycle substrate can be used to generate energy and anabolic substrates such as amino acids or for fatty acid synthesis via reverse carboxylation. A mechanism has been identified by which *TXNIP* is differentially regulated in response to glucose or glutamine [94]. This involves the recruitment of MondoA–Mlx heterodimers to the *TXNIP* promoter where they associate with one or more histone acetyltransferases, which activate the gene, inhibit glucose uptake and temper the Warburg effect. Both glucose and glutamine alter this interaction such that MondoA–Mlx heterodimers interact with histone deacetylases, thereby mitigating *TXNIP* induction and allowing higher levels of glucose uptake. While this model is appealing, it is plausible that other Oxphos intermediates also regulate *TXNIP* expression as evidenced by the fact that inhibiting Oxphos actually inhibits *TXNIP* expression rather than induces it [66]. Other factors that activate *TXNIP* and/or *ARRDC4* include lactate and non-glucose hexoses, whereas serum stimulation of quiescent diploid fibroblasts inhibits *TXNIP* [32,62,92,377].

In many tumors, Myc up-regulates not only the Warburg effect but mitochondrial biogenesis, Oxphos and glutaminolysis as well [41,78,389,392,393]. In doing so, Myc dysregulation has been shown to indirectly impact the activity of MondoA and its activation in several distinct ways. First, by strongly stimulating the uptake of glucose and glycolysis, Myc provides the key metabolites needed to promote MondoA’s nuclear translocation and transcriptional activation. Second, the additional transcriptional induction of glycolytic genes provided by MondoA reinforces those which are already Myc-responsive such as *PLKR* [76,77,313]. Third, Myc-mediated promotion of glutaminolysis activates the pathway by which MondoA suppresses *TXNIP* and *ARRDC4,* thereby contributing to glycolysis by relieving the block to glucose uptake. Finally, the well-known induction of *LDHA* by Myc [394] provides the lactate needed to further induce transcription of both *TXNIP* and *ARRC4* genes. Together, these observations reinforce one of the central themes of this review, namely that any given member of the Extended Myc Network does not function alone but rather is dependent on the well-integrated and cooperative activities of its other members.

## 12. Mlx

Originally identified in yeast two-hybrid screens that used Mxd1 or Mnt as baits, Mlx’s bHLH-ZIP domain bears significant structural and functional homology to Max [74,285,395], hence the designation Max-like protein X (Mlx). The expression pattern of Mlx, both in adult and embryonic tissues, is quite broad and also resembles that of Max [45]. The Mlx protein is stable, forms avid E-box-binding heterodimers in association with Mxd1 and, also like Max, binds DNA as a homodimer albeit with lower affinity than its heterodimeric forms [285,395]. Mlx also heterodimerizes with Mxd4, Mnt, ChREBP and MondoA [45,74,285,395] (Figure 1).

Mlx’s collaboration with ChREBP and MondoA may not be the only means by which it reprograms metabolism and cell behavior, particularly with regard to the maintenance of the undifferentiated state that typifies most cancers [396]. For example, MondoA expression is particularly prominent in skeletal muscle, where high levels of glucose are necessary for myogenesis and where glycolytic demand and activity are high [374,397]. In a study focused primarily on the murine C2C12 myogenic cell line, Mlx over-expression induced the expression of several soluble myokines, such as insulin-like growth factor 2, that promote skeletal muscle differentiation [398,399]. This induction was direct as indicated by the documentation of Mlx being localized to ChoREs in the proximal promoters of some of these genes and an associated increase in histone H4 acetylation at nearby sites. Surprisingly, despite Mlx also directly activating *Txnip* and *Arrdc4* in these cells in a glucose-dependent manner, it did not induce any of the usual glycolytic gene repertoire. This indicated that the function of Mlx in skeletal muscle (in collaboration with MondoA) was quite different from its function in liver where its major heterodimeric partner is ChREBP. Interestingly the ChoREs within the *Txnip* promoter, which were initially occupied by Myc, were displaced by over-expressed Mlx. Given that the ectopic expression of Myc can block myogenic differentiation [400], these findings suggested an attractive model by which myogenesis was regulated by the balance between the Myc and Mlx Networks. In agreement with this, C2C12 myoblasts with enforced expression of Mlx underwent fusion-associated differentiation more rapidly than control cells, whereas myoblasts with a knockdown of Mlx showed evidence of a differentiation block. It is tempting to consider that the loss of *Mlx* expression, whether by mutation, gene deletion or promoter silencing, inactivates its TS-like function and tips the balance in favor of the Myc Network, thereby maintaining both proliferation and the undifferentiated state.

A role for Mlx in overseeing normal hepatocyte proliferation has recently been described in a study that again employed the *Fah^−/−^* mouse model of hereditary tyrosinemia [41,76,78,371]. Donor hepatocytes expressed an albumin gene promoter-driven Cre-estrogen receptor (CreER) fusion protein that allowed for the tamoxifen-deriven excisional inactivation of the ‘floxed’ endogenous *Mlx* locus 3–4 months after birth. Anticipating from prior work that these *Mlx*KO hepatocytes would be at a proliferative disadvantage [41,78], they were administered intrasplenically to recipient *Fah*^−/−^ mice together with WT hepatocytes at a ~6:1 ratio. The donor population was assessed 24–28 weeks later in the reconstituted liver and was found to be >95% WT. Thus, despite their initially large numerical advantage, *Mlx*^−/−^ hepatocytes were nevertheless heavily outcompeted.

The above findings were extended by performing similar competitive repopulation studies using a double knockout (DKO) population of *Myc*^−/−^ x *Mlx*^−/−^ hepatocytes that were delivered to *Fah*^−/−^ recipients together with WT cells at a ~10:1 ratio [76]. An end-of-study reassessment showed that WT hepatocytes now comprised >97% of the donor population. Suspecting that these cells were even more deficient than the previously studied *Mlx*^−/−^ hepatocytes, a third round of competitive repopulation was performed with a 1:1 ratio of *Mlx*^−/−^ and DKO hepatocytes. Despite their profound proliferative disadvantage relative to WT cells noted above, *Mlx*^−/−^ cells still possessed a clear growth advantage over DKO cells, with the former population comprising nearly 95% of the retrieved donor pool. Taken together, these studies as well as those reported previously and discussed above, indicated that the regenerative potential of normal hepatocytes declines as the Extended Myc Network is progressively dismantled (WT=*Myc*^−/−^
*>Chrebp*^−/−^
*>Myc*^−/−^
*x Chrebp*^−/−^
*>Myc*^−/−^
*x Mlx*^−/−^*)* [41,76,78].

Gene expression profiling in *Mlx*^−/−^ and DKO livers documented dysregulation of many of the same pathways previously described as being altered in *Myc*^−/−^*, Chrebp*^−/−^ and *Myc*^−/−^
*x Chrebp*^−/−^ livers [76]. Most notably, these included gene sets with roles in ribosomal structure and function, mRNA processing and translational, energy metabolism and mitochondrial structure and function, including transcripts encoding most mitochondrial ribosomal subunits. All these gene sets were down-regulated in their respective knockout livers, with particularly robust suppression being seen in DKO livers. This correlated quite well with the above-described competitive repopulation studies and indicated that the most severe proliferative defects were associated with the greatest degree of suppression of target genes that were involved in the processes (translation and energy production) most needed to sustain growth and proliferation. As previously described in comparing *Myc*^−/−^ and *Chrebp*^−/−^ livers [41], significant overlap in gene expression changes was observed, providing support for cross-regulation of one another’s transcriptional targets by each arm of the Extended Myc Network. A carefully-performed analysis of ChIP-seq results from a HepG2 hepatocellular carcinoma (HCC) cell line indicated that many direct Myc and Mlx target genes bound to common E-boxes and ChoREs.

Despite the inherent proliferative defects of their hepatocytes, over one-third of the above *Mlx*^−/−^ and DKO mice developed multiple small liver tumors by 14–16 mos. of age compared to none in similarly-aged WT, *Myc*^−/−^*, Chrebp*^−/−^ and *Myc*^−/−^
*x Chrebp*^−/−^ mice [76]. Histologically, these tumors were well-differentiated and/or myxoid-type adenomas containing many Ki67-positive cells and rare foci of hepatocellular carcinoma. Their lack of expression of the deleted gene(s) indicated that they did not originate from a minority population of cells that had failed to excise the target and therefore maintained a replicative advantage. Whole-transcriptome profiling of these tumors confirmed their lack of expression of Myc and/or Mlx while showing that they were readily distinguishable from normal liver and previously characterized HBs [41,78,117,346]. The fact that similar tumors had not been previously observed in *Chrebp*^−/−^ mice strongly suggested that MondoA serves a redundant function in protecting against neoplastic conversion and that the functional inactivation of both Chrebp and MondoA were needed for this feature to be revealed. Because these studies were terminated when mice reached the age of 14–16 months and because gene knockouts were confined to hepatocytes, the actual penetrance of the tumor phenotype over the entire lifespan of the animals remains to be determined as does the question of whether *Mlx* is a more general TSG in other tissues and, if so, what types of tumors are associated with its loss. Nonetheless, these studies showed *Mlx* to be a potent TSG in the context of an otherwise normal genetic background and in the absence of any predisposing factors.

Histologically, the adenomas originating in *Mlx*^−/−^ and DKO livers closely resemble their human counterparts [401,402]. However, they differed in several respects that included their multi-focality and high-level expression of Ki67 (~20–30% of cells), both of which are rare features of human hepatic adenomas [403]. *Mlx*^−/−^ and DKO adenomas also dysregulated 15 transcripts that had been previously shown to be associated with inferior survival in HBs and over a dozen adult cancers [213]. These characteristics, the occasional tendency of these adenomas to evolve into HCCs and their unique transcriptional profiles [213], suggests that they occupy a transitional state between benign and malignant neoplasms and may take on more of the latter’s features as mice age.

It is also important to consider the possibility that the observed hepatic neoplasms were related to the pronounced non-alcoholic fatty liver disease (NAFLD) that is associated with the loss of individual Extended Myc Network members and appears earlier in DKO animals [41,76,78]. NAFLD is a known risk factor for the development of both adenomas and HCC [403,404,405]. However, while potentially contributing to the evolution of adenomas, it seems unlikely that NAFLD per se was directly causative given that adenomas have not been observed in the livers of *Myc*^−/−^*, Chrebp*^−/−^
*or Myc*^−/−^
*x Chrebp*^−/−^ mice [76] which also developed equally pronounced NAFLD [76].

The seemingly paradoxical role played by Mlx in promoting normal hepatocyte proliferation while simultaneously serving as a (benign) TSG [76] has direct parallels with Max, the central player in the Myc Network (Figure 1). The most likely explanation for these opposed yet compatible behaviors is that The Mlx Network, like that of the Myc Network, oversees (along with other factors) both normal and neoplastic proliferation, with ChREBP and MondoA overseeing a set of genes that collectively supports proliferation and Mxd1, Mxd4 and Mnt overseeing a set of genes that suppress tumorigenesis (Figure 1). The loss of both functions in response to *Mlx* knockout favors the default outcome of neoplasia in ways that might be influenced by the Myc Network as well as by other, as yet unknown players, given that *Mlx* continue to serve as a TSG even when the Myc Network is concurrently activated [76].

## 13. Conclusions

This review, along with other studies, has emphasized both the complexity and flexibility of the Extended Myc Network in promoting normal and neoplastic cell growth largely by regulating ribosomal biogenesis and translation, metabolic and energy-generating pathways, cell cycle progression and other key pathways [25,26,31,38,41,42,50,70,76,78,86,88,372,391,392,393,406]. Yet the unavoidable conclusion is that most members of this highly interconnected network function more as TSGs than as oncoproteins. Indeed, the only ones with indisputable oncogenic functions [407] are *MYC* and its paralogs, while the tumor facilitators *CHREBP* and *MONDOA* promote growth and proliferation without being directly oncogenic [41,76,78].

Additional layers of complexity among members of the Extended Myc Network have recently been identified with machine learning-based dimensionality reduction techniques such as t-SNE or UMAP that simultaneously compared the relationships among all Extended Myc Network member transcripts [408,409]. These studies showed most cancers to be comprised of 2–5 distinct clusters of Extended Myc Network transcripts that had prognostic value beyond that afforded by standard whole-transcriptome profiling or the examination of single transcripts such as those depicted in Figure 3 and Figure 4. Even favorable and unfavorable prognostic groups, initially identified by whole-transcriptome profiling, could be further subdivided based on the clustering patterns of Extended Myc Network transcripts and vice versa thus showing the two methods to be complementary. This approach proved useful in 10 different cancer types, comprising over one-third of the entire TCGA population. Moreover, such clustering was not confined to Extended Myc Network transcripts. Indeed, it had been previously demonstrated with human ribosomal protein transcripts whose patterns also correlated with molecular and pathological features and long-term survival in some cancers [113,410]. Eventually, the approach was applied to a collection of 212 transcripts representing eleven additional cancer-related pathways such as those involved in Wnt and Notch signaling, purine and pyrimidine biosynthesis and the TCA cycle. Clustering was predictive of survival in 30 of the 34 cancers, or 91.4% of all tumors in TCGA [408].

This review has raised a number of questions that will be important to address in future work. Among these are the degree to which direct Myc Network and Mlx Network target genes are co-regulated and the extent to which their binding sites are shared [76]. What are the gene expression consequences when targets are bound by different heterodimeric combinations of factors versus only a single type and to what extent is such differential binding even possible? It is already known that Myc–Max targets overlap Mxd–Max targets but how are these sites selected, what determines site selection and how and why does it change under different circumstances? The simplicity of E-boxes and the degeneracy of ChoREs imply much more potential than actual binding sites. What determines binding in the first place and why do particular E-boxes specifically bind Myc, Mxds or Mlx? How is the binding to these sites altered in different tissues or when a tissue undergoes transformation due, for example to Myc over-expression or the loss of Max or Mlx? What is the role of epigenetic modification in altering binding site affinities [168]? It seems likely that many E-boxes and ChoREs represent ‘latent’ elements with low binding affinities and little functional relevance until Myc is over-expressed or dysregulated during tumorigenesis. Finally, since Myc and Mlx binding sites can be shared, is it possible that the ChREBP/MondoA metabolite responsiveness of at least some genes can be “acquired” in a manner that reflects the abundance of these factors, their intrinsic affinity for their binding sites, the competition of Myc Network members for the site and the intracellular concentration of enabling metabolites? The contribution of genes associated with these sites is likely to provide significant insights into the pathogenesis of many tumor types that are initiated and/or maintained by high Myc levels. Finally, what is the role in tumor suppression for genes bound exclusively by Mxd members versus those in which these factors can be displaced by Myc, ChREBP and MondoA? Having filled in the basic components and workings of the Extended Myc Network, we are now positioned to begin to address these more subtle questions and thus to reveal fundamental aspects of their oncogenic and TS functions. Exciting times lie ahead!

## Figures and Tables

**Figure 1 cells-11-00747-f001:**
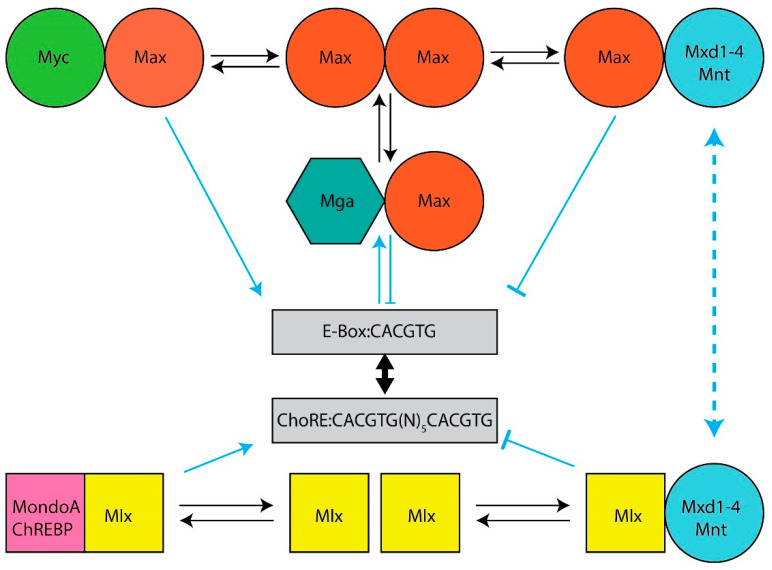
**The Extended Myc Network.** Top: The Myc Network. Myc–Max heterodimers bind to cognate E-boxes in target genes. They then recruit chromatin-modifying enzymes such as histone acetyl transferases and transcription co-factors, which mediate the histone acetylation and demethylation that permit subsequent increases in RNA polymerase II (Pol II) binding, the relief of Pol II pausing and read-through transcription [14,15,16,17,18]. DNA binding by Max homodimers is prevented by inhibitory N-terminal phosphorylation [19,20]. Max also lacks a TAD that is needed for chromatin remodeling and transcriptional activation [13,16,21]. Transcriptional repression of Myc-activated genes is mediated by six “Mxd proteins” (Mxd1–4, Mnt and Mga) whose levels of expression are variably tissue, development, age and cell cycle specific [22,23,24,25,26,27]. These compete with Myc–Max heterodimers for E-boxes, recruit mSin3, histone deacetylases, methyltransferases and complexes that mediate more direct transcriptional repression [28,29,30]. Bottom: The Mlx Network. The Myc-like factors MondoA and ChREBP heterodimerize with Mlx (which, unlike Max, can homodimerize, albeit weakly), bind certain metabolites such as glucose, glucose-6-phosphate, lactate and adenosine, and translocate from the cytoplasm to the nucleus where they bind to target genes containing both ChoREs and E-boxes although the size of this repertoire is smaller than that of Myc targets [31,32,33,34,35,36,37,38,39,40,41,42]. The negative regulatory arm of the Mlx Network employs some of the same repressive strategies utilized by the Myc Network with some distinct exceptions.

**Figure 2 cells-11-00747-f002:**
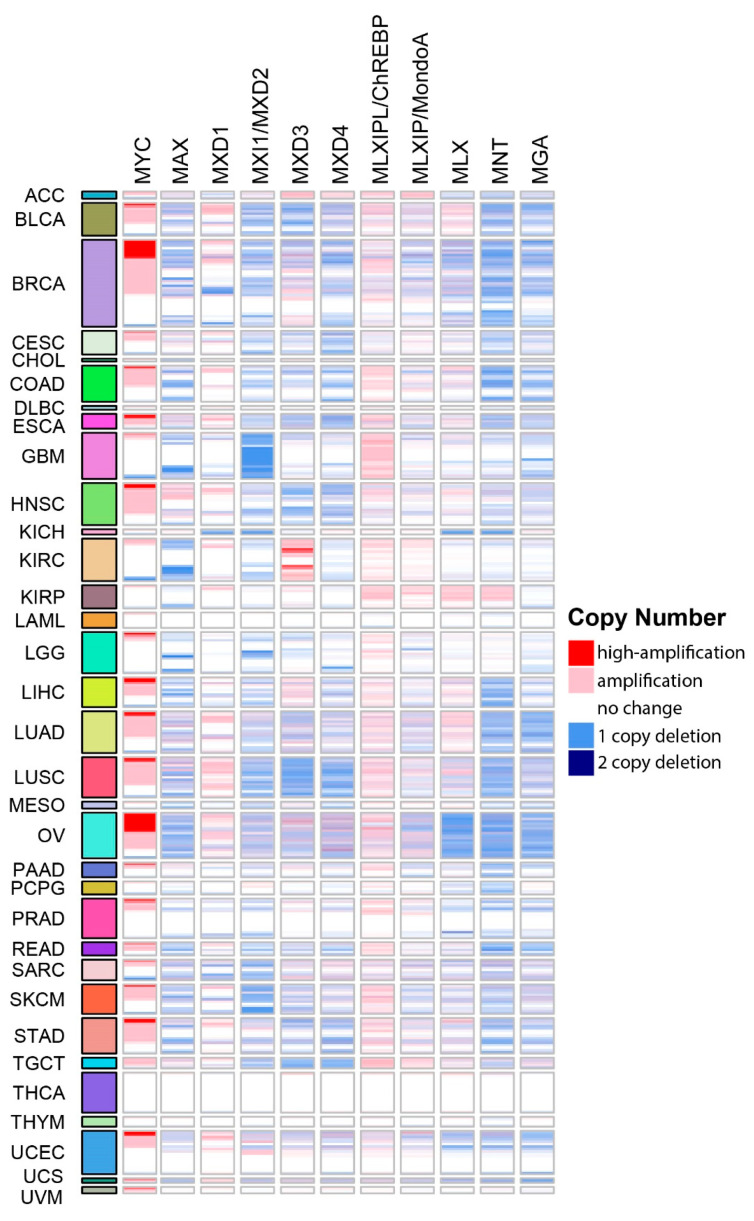
**CNVs among individual members of the Extended Myc Network.** Gene-level copy number (gistic2_thresholded) [111] data were downloaded from the TCGA Pan-Cancer (PANCAN) database (https://xenabrowser.net/datapages/?cohort=TCGA%20Pan-Cancer %20(PANCAN), accessed on 15 November 2021) Heatmaps were drawn using ComplexHeatmap R package [112]. Tumor type abbreviations: ACC: adrenocortical carcinoma; BLCA: bladder urothelial carcinoma; BRCA: breast invasive carcinoma; CESC: cervical squamous cell carcinoma/endocervical adenocarcinoma; CHOL: cholangiocarcinoma; COAD: colon adenocarcinoma; DLBC: diffuse large B-cell lymphoma; ESCA: esophageal carcinoma; GBM: glioblastoma multiforme; HNSC: head and neck squamous cell carcinoma; KICH: kidney chromophobe carcinoma; KIRC: kidney clear cell carcinoma; KIRP: kidney renal papillary cell carcinoma; LAML: acute myeloid leukemia; LGG: lower-grade glioma; LIHC: hepatocellular carcinoma; LUAD: lung adenocarcinoma; LUSC: lung squamous cell carcinoma; MESO: mesothelioma; OV: ovarian serous cystadenocarcinoma; PAAD: pancreatic adenocarcinoma; PCPG: pheochromocytoma/paraganglioneuroma; PRAD: prostate adenocarcinoma; READ: rectal adenocarcinoma; SARC: sarcoma; SKCM: skin cutaneous melanoma; STAD: stomach adenocarcinoma; TGCT: testicular germ cell tumor; THCA: thyroid carcinoma; THYM: thymoma; UCEC: uterine corpus endometrial carcinoma; UCS: uterine carcinosarcoma; UVM: uveal melanoma.

**Figure 3 cells-11-00747-f003:**
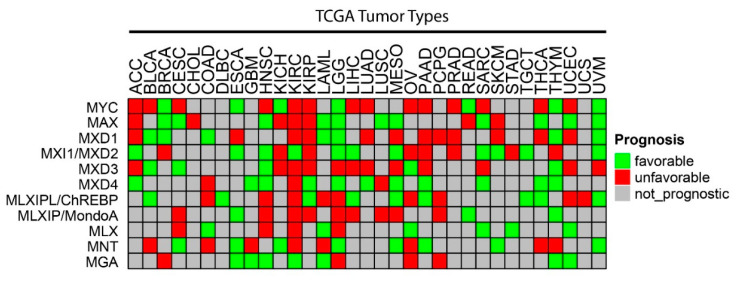
**Correlations between survival and expression of members of the Extended Myc Network.** Based on RNAseq results from TCGA, the expression of each Extended Myc Network member was examined in the indicated tumors from the same database. Tumors were divided into subsets with overall favorable or unfavorable survival. The numbers in each of these subsets were based on the levels of transcript expression that provided the most significant survival differences. The “survival” and “survminer” R packages were used for survival analysis. FPKM value ranks were used to classify individuals into two groups by 50 series cutoffs ranging from 10 to 90%. Survivorship was examined by Kaplan–Meier survival estimators, and the survival outcomes of the two groups were compared by log-rank tests. Curves were based on the FPKM cutoffs that yielded maximal survival difference between the two groups with the lowest log-rank *p*-value. See Legend to Figure 2 for tumor type abbreviations.

**Figure 4 cells-11-00747-f004:**
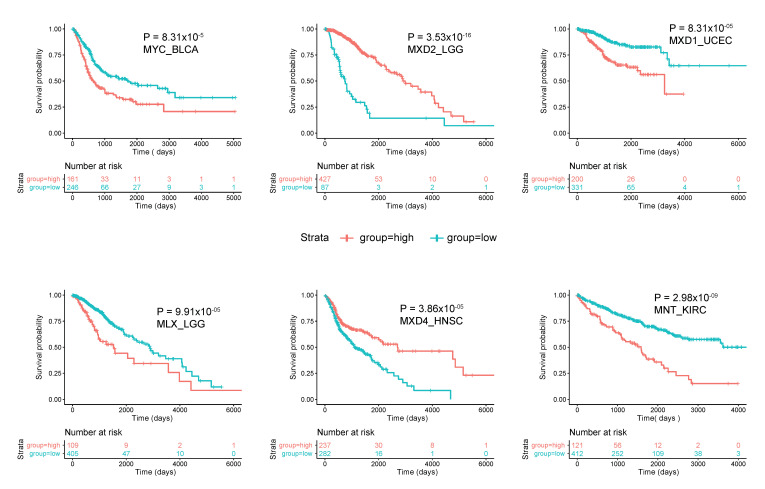
**Correlations between survival and expression of select members of the Extended Myc Network.** Tumors were chosen from those in TCGA and further subdivided as described in Figure 3. All curves were generated using the “survival” and “survminer” R packages. The number of individuals in the favorable and unfavorable survival groups which provided the greatest significance in survival is depicted in the “0 time point” beneath each survival curve.

**Figure 5 cells-11-00747-f005:**
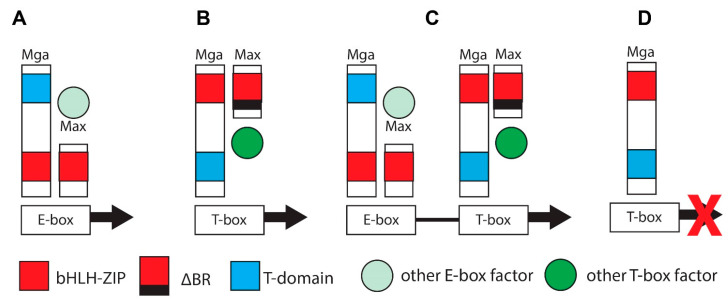
**DNA binding specificity and transcriptional activation by Mga.** (**A**). E-box binding. DNA binding is mediated by the cooperative interaction between the bHLH-ZIP domains of Mga and Max. This requires an intact basic domain of Max, and presumably Mga as well. Transcriptional activation is indicated by the block arrow. (**B**). T-box binding. This is mediated via the T domain of Mga. Transcriptional activation requires dimerization with Max but does not require Max’s basic domain since a Max BR mutant lacking this domain functions in the same way as intact Max. Max’s role is unclear but may be involved, either directly or indirectly in the recruitment of other factors that are needed for chromatin modification and/or the formation of an active transcriptional activation complex. The factors are different from those recruited by Myc–Max heterodimers and are referred to here as ‘T-box factors’. (**C**). Cooperative dual E-box and T-box binding. Mga binds both sites in the same manner and with the same requirements as binding to individual sites. (**D**). In the absence of Max and other necessary co-factors, Mga still binds its T-box sites but acts as a transcriptional repressor.

## Data Availability

Not applicable.

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
