# Peer review of "Normal and Neoplastic Growth Suppression by the Extended Myc Network"

_cells, 2022, doi:10.3390/cells11040747_

Round 1

Reviewer 1 Report

In this manuscript, the authors provide a review of the extended MYC network, with a particular focus on the cross-talk between the MYC-MAX-MXD and ChreBP-Mlx-MondoA arms as well as the association of this extended MYC network with cancer. This is a well-written, extensive review wherein the authors have described each individual protein in this network and outlined their role in cancer based on current literature.

Specific comments:

  1. The authors refer to MGA as one of the MXD proteins. However, in the literature, it seems that MGA is considered separate from them. Therefore, could the authors please provide reference(s) for where MGA is classified as an MXD protein?

  1. Lines 83-84 “This “Mlx network” also comprised of BHLH-ZIP factors, contained the ‘positively-acting’ Myc-like proteins…”: Strictly speaking, the words ‘positively acting’ do not explain much here. Are they ‘positively-acting’ in terms of transcriptional activation (presumably) or phenotypic outcomes? Please substitute these words accordingly.

  1. Figure 2: Please label the figure with full names of the tumor types instead of abbreviations.

  1. Lines 171-172 “in the case of low-grade gliomas (LGGs) with all 13 members, although not necessarily the same way.”: First, Figure 3 shows that this is the case for KIRC as well and not just LGGs alone. Therefore, please correct this statement to include KIRC. Second, the authors refer to 13 members when there are only 11. Please, correct.

  1. Lines 211-213 “Yet, the recurrent loss of Max in pheochromocytomas, paraganglioneuromas and several other tumors supports its role as a potent TS [106-112].”: In addition to the references cited, the authors’ own data (Figure 2) supports this statement. Therefore, please cite Figure 2 here.

  1. Lines 252-253: The authors jump from tumor-promoting to tumor-suppressing role of MYC without an introduction to the fact that MYC could also act as a potential tumor suppressor. This feels a little abrupt. Please, add sentence to introduce this concept. (Please, also take into account the next 2 comments regarding Myc being a tumor suppressor vs a less potent oncogene)

  1. Lines 253-264: In this paragraph, the authors discuss the optimization of Myc-mediated transcriptional activation in the presence of HectH9, a protein that is inhibited by Miz1. From this, they draw the conclusion that “MYC toggles between two states that alternatively activate or suppress transcription of the same target gene.” However, the evidence summarized does not indicate any role of MYC in suppressing transcription in the absence of HectH9. It seems that MYC is merely ‘less active’ as an oncogene in the absence of Hect9. Therefore, this does not qualify it as a tumor suppressor in this context. Its activity is merely regulated by HectH9-Miz1. Therefore, please correct.

  1. Lines 265-268 “Other post-translational modifications that could contribute to Myc’s role in tumor suppression occur via two distinct mechanisms involving the phosphorylation of T358, S373 and T400 by the Ser/Thr kinase Pak2. These alterations lead to a loss of Myc-Max heterodimerization and reduced target affinity [130]”: again, do these post-translational modifications merely dampen the tumorigenic potential of MYC or do they confer to it a tumor-suppressive role?

  1. Lines 271-276: The molecular detail explaining the interaction of RARa with phosphorylated and unphosphorylated Myc is not necessary. Only mentioning that ‘phosphorylated MYC activates RARa-mediated transcription whereas unphosphorylated MYC inhibits it’ would suffice and would help keep the review concise. Therefore, please make this change.

  1. Line 305: Again, the switch from inhibiting Myc-Max association for therapeutic purposes to evidence for Max’s potential as a TS is rather abrupt. Please, include a sentence for this transition.

  1. Lines 311-320: The following phrases can be removed for conciseness:
  • “derived from adrenal chromaffin cells or extra-adrenal sympathetic and para sympathetic tissues of neural crest origin [150]”
  • “particularly those adrenal tumors that are bilateral and non-adrenal tumors that are multi-focal.”
  • “with the most common being those encoding any of the four subunits of the succinate dehydrogenase complex of the electron transport chain, otherwise known as complex II (SDHA, SDHB, SDHC and SDHD) [107-109, 112, 152].”

  1. Lines 358-359: “SCLC in humans is often associated with TP53 and RB1 loss/mutation and amplification of MYC or its paralogs MYCN and MYCL (Fig. 2)”. Figure 2 does not seem to include SCLC. Please, correct.

  1. Lines 447-483: Again, for conciseness, these 2 paragraphs can be summarized in a few sentences stating which proteins co-localize (and at which sites) and which don’t. Additionally, the authors don’t mention what this differential spatial-distribution means in the context of cancer, which is the focus of the review. Have alterations in this distribution or co-localization pattern been observed in any cancers/ cancer models?

  1. Lines 484-503: Again, the authors mention no link of this regulation by spatial distribution to cancer. Please, correct.

  1. Lines 520-536: This paragraph discusses the role of Mxd1 in promoting chemotherapeutic drug resistance, which is a pro-tumorigenic feature. The authors, however, refer to it as a tumor suppressive role of Mxd1, which is contradictory. Please, correct. The concept of “transcriptional repression” should not be confounded with “tumor suppression”.

  1. Line 618 “thus being better more efficient at suppressing…”. Please remove the word ‘better’.

  1. Lines 745-747 “Among all tumors, the mean levels of…”: What are the authors referring to as ‘all tumors’? Are these primary LGGs or primary and recurrent LGGs combined? Please, clarify.

  1. Lines 824-828 “Mnt has slightly different sequence-specific DNA binding preferences compared to Myc and Mxd1-4 [26,281]. Like them however, it contains an mSin3 A/B-interacting SID domain that is necessary for the HDAC-mediated transcriptional repression of E-box containing target genes and the inhibition of MYC+RAS-mediated transformation.” The underlined part of the text only refers to the features of Mxd1-4 but not MYC. However, the phrase ‘Like them’ includes all MYC and Mxd1-4. Please, correct.

  1. Lines 859-860 “Interestingly, rather than suppressing proliferation like Mxd1-4 or as it does in fibroblasts [26], Mnt was shown…”. According to the section on Mxd3, it does not suppress proliferation but promotes it instead. Therefore, please correct.

  1. Lines 919-948: These two paragraphs compare and contrast the role of Mnt and Myc in two different models. In addition to being distinct cancer models, the first one investigates the effect of heterozygous Mnt knockout whereas the second one looks at homozygous knockout. Could the authors please discuss the potential effect of these differences on the outcomes of these studies?

  1. Paragraph 963-977: Based on the findings summarized in this section, it seems that Mnt LOH (sometimes alongside MYC amplification) was pro-tumorigenic, implying Mnt’s role as a TSG in this context. However, in the 2 paragraphs in lines 910-937, the opposite seems to be the case despite the fact that they all look at T-cell-based models. How do the authors explain this contradiction? Please discuss in the text.

  1. Lines 1001-1003 “Collectively, these studies suggest that the higher expression of Mnt associated with EMT in breast cancer actually serves to temper proliferation and thus serves as a TS.” : While Mnt serving as a TS in the context of proliferation is correct, this does not apply to the context of EMT, in which case, it serves more as a pro-tumorigenic factor. Therefore, could the authors please weigh and comment on both the aspects instead of being restricted to proliferation alone?

  1. Line 1255 “However, the interaction between MCR IV and 14-3-3 requires neither glucose…” : In line 1248, the authors state that 14-3-3 interacts with MCR III instead. Which one is correct? Also, please be consistent between roman and Arabic numbers when mentioning MCRs.

  1. Lines 1350-1364: The paragraph begins by stating that “high levels of MondoA expression were found in leukemic blasts…”, implying its role as an oncogene. However, in lines 1357-1359, the authors refer to a study that showed that knockdown of MondoA in primary ALL blasts resulted in downregulation of genes associated with better therapeutic outcomes and survival. Therefore, it seems that despite being oncogenic, MondoA still confers a survival and therapeutic benefit. Could the authors please comment on that?

  1. Lines 1449-1451 “It is tempting to consider that that the loss of Mlx expression whether by mutation, gene deletion or promoter silencing could contribute to the maintenance of the undifferentiated state and thus serve as a TS.” From this statement, it is unclear if Mlx itself or its loss of expression serves as a TS. Please rephrase to clarify.

  1. Lines 1451-1475: The findings summarized in these paragraphs point to a pro-proliferative role of Mlx in normal hepatocytes. However, the findings in lines 1491-1510 suggest a tumor suppressive function. How do the authors explain these contrasting observations? Please comment.

  1. A majority of the Greek letters are missing from the text. Instead, there is just a blank space where the symbols should be (for example, TGF-b, HIF-a). Please, revise and correct.

  1. There are a few typos throughout the text. Please, revise and correct.

Author Response

Reviewer 1

We wish to particularly thank this Reviewer whose thorough and well-considered comments indicated that he/she carefully read the manuscript in its entirety. This allowed us to clarify a number of points, correct several inadvertent mistakes and provide better overall organization.

  1. Comment 1: The authors refer to MGA as one of the MXD proteins. However, in the literature, it seems that MGA is considered separate from them. Therefore, could the authors please provide reference(s) for where MGA is classified as an MXD protein?

Response: The Reviewer is correct that Mga (and Mnt as well) are structurally and functionally distinct from the more closely-related Mxds1-4 as we discuss throughout the review.  We stated in the manuscript that these six proteins, with transcriptionally repressive functions, would be “…referred to collectively as ‘Mxd proteins’ ”. This was done to avoid potentially confusing and more cumbersome nomenclature (“Mxd1-4/Mnt/Mga”) each time the proteins were referred to collectively. The group has been previously described in this manner by Diolaiti et al (Biochim Biophys Acta 2015.1849:484-500) and we cited this paper in our Bibliography.  In the revision, the above sentence has been changed to read  “…referred to collectively as “Mxd proteins, despite their structural and functional differences”.

  1. Comment 2: Lines 83-84 “This “Mlx network” also comprised of BHLH-ZIP factors, contained the ‘positively-acting’ Myc-like proteins…Strictly speaking, the words ‘positively acting’ do not explain much here.”

Response: The term “positively-acting” has been removed. The reference to ChREBP and MondoA as “Myc-like”, Fig. 2 and the ensuing discussion should be sufficient to inform the reader that these factors up-regulate their targets.

  1. Comment 3: Figure 2: Please label the figure with full names of the tumor types instead of abbreviations.

Response: The abbreviations are the accepted TCGA nomenclature. However, full names have now been added to the legend (adding them to the figure itself detracted from its appearance).

  1. Comment: Lines 171-172 “in the case of low-grade gliomas (LGGs) with all 13 members, although not necessarily the same way.”: First, Figure 3 shows that this is the case for KIRC as well and not just LGGs alone. Therefore, please correct this statement to include KIRC. Second, the authors refer to 13 members when there are only 11.

Response: Thank you for pointing this out.  The errors have been corrected.

  1. Comment: Lines 211-213 “Yet, the recurrent loss of Max in pheochromocytomas, paraganglioneuromas and several other tumors supports its role as a potent TS [106-112].”: In addition to the references cited, the authors’ own data (Figure 2) supports this statement. Therefore, please cite Figure 2 here.

Response: The reference to Fig. 2 has been now been added.

  1. Comment: Lines 252-253: The authors jump from tumor-promoting to tumor-suppressing role of MYC without an introduction to the fact that MYC could also act as a potential tumor suppressor. This feels a little abrupt. Please, add sentence to introduce this concept. (Please, also take into account the next 2 comments regarding Myc being a tumor suppressor vs a less potent oncogene).

Response: We have now altered the introductory sentence of this paragraph to read: “Despite its long and storied storied history as an oncogene, Myc may also play a role in tumor suppression.  A novel means by which this might occur has been suggested by studies…”

  1. Comment(s): Lines 253-264: In this paragraph, the authors discuss the optimization of Myc-mediated transcriptional activation in the presence of HectH9, a protein that is inhibited by Miz1. From this, they draw the conclusion that “MYC toggles between two states that alternatively activate or suppress transcription of the same target gene.” However, the evidence summarized does not indicate any role of MYC in suppressingtranscription in the absence of HectH9. It seems that MYC is merely ‘less active’ as an oncogene in the absence of Hect9. Therefore, this does not qualify it as a tumor suppressor in this context. Its activity is merely regulated by HectH9-Miz1. Therefore, please correct.

Response: The Reviewer is technically correct on this point. The original paper we cited and that first identified this mechanism (Adhikary et al. Cell 2005.123:409-421) stated in its Abstract: “…site-specific ubiquitination regulates the switch between an activating and a repressive state of the Myc protein” (italics ours), which is arguably an over-statement. It is probably more accurate to interpret this as the reviewer suggests, namely that Myc toggles between two states representing high and low transcriptional activation potential rather than a form of Myc that actively engages in tumor suppression in the classical sense. We certainly do not disagree with this interpretation. Adhikary et al (op cit) did suggest that HectH9 could potentially represent a therapeutic target that would prevent the Ub-mediated activation of Myc. In the revised manuscript, we suggest that the under-ubiquitinated form of Myc is not only less active but may also behave in a dominant negative manner. We have modified this section of the revised manuscript accordingly.

  1. Comment: Re other post-translational modifications mediated by the Ser/Thr kinase Pak2.

Response: We believe that the response to point 7 above is of a similar nature and we again do not disagree with the Reviewer.

  1. Comment: Lines 271-276: The molecular detail explaining the interaction of RARa with phosphorylated and unphosphorylated Myc is not necessary.

Response: This section has been shortened as requested.

  1. Comments: Line 305: Again, the switch from inhibiting Myc-Max association for therapeutic purposes to evidence for Max’s potential as a TS is rather abrupt. Please, include a sentence for this transition.

Response: A sentence that connects these two lines of thought has now been added as suggested.

  1. Comment: Lines 311-320: The following phrases can be removed for conciseness: “derived from adrenal chromaffin cells or extra-adrenal sympathetic and para sympathetic tissues of neural crest origin”… “particularly those adrenal tumors that are bilateral and non-adrenal tumors that are multi-focal.”…”with the most common being those encoding any of the four subunits of the succinate dehydrogenase complex of the electron transport chain, otherwise known as complex II (SDHA, SDHB, SDHC and SDHD).”

Response: The suggested deletions have been made

  1. Comment: Lines 358-359: “SCLC in humans is often associated with TP53and RB1loss/mutation and amplification of MYC or its paralogs MYCN and MYCL (Fig. 2) (161-163)”. Figure 2 does not seem to include SCLC. Please, correct.

Response: SCLC is not one of the tumor types that was analyzed by the TCGA Consortium and our reference to it as being represented in Fig. 2 was inadvertent. Refs. 161-163 are instead used to support this statement.  Thank you for pointing out this oversight.

  1. Comment: Lines 447-483: Again, for conciseness, these 2 paragraphs can be summarized in a few sentences stating which proteins co-localize (and at which sites) and which don’t. Additionally, the authors don’t mention what this differential spatial-distribution means in the context of cancer, which is the focus of the review. Have alterations in this distribution or co-localization pattern been observed in any cancers/ cancer models?

Response: As suggested, we have reduced the length of this section by >50%. Although our results have been reproduced by others in various contexts, neither we nor they have determined the significance of these findings for cancer (Suzuki T, et al. J Reprod Dev. 2009.55:491-5; Osborne CS, et al. PLoS Biol. 2007.5:e192; Su Y, et al. Genes Dev. 2018. 32:1398-1419). We did note in our own work (Yin et al. Oncogene 2001.20:4650–64) that the regions responsible for nuclear localization and speckling are separable and that deletion of the latter domain in Mxd2 abrogated its repressive function while allowing it to remain confinred to the nucleus. This would presumably impact the protein’s function in the context of cancer.

  1. Comment: Lines 484-503: Again, the authors mention no link of this regulation by spatial distribution to cancer. Please, correct.

Response: The above discussion (Comment 13) centered around the presumptive role of Mxd1 in regulating the nucleolar transcription of rRNA genes, which are well-known Myc targets whose expression correlates with rapid tumor growth as we mentioned in the review and have shown in our own work (Grandori C, et al. Nat Cell Biol. 2005.7:311-8; Wang H, et al. J Biol Chem. 2018.293:14740-57.) No descriptions of intranucleolar spatial segregation among members of the Network were reported by these workers.

  1. Comment: Lines 520-536: This paragraph discusses the role of Mxd1 in promoting chemotherapeutic drug resistance, which is a pro-tumorigenic feature. The authors, however, refer to it as a tumor suppressive role of Mxd1, which is contradictory. Please, correct. The concept of “transcriptional repression” should not be confounded with “tumor suppression”.

Response: We thank the Reviewer for pointing this out. This section should not have been included in a discussion of Mxd1’s putative role as a TS and it has therefore been removed. 

  1. Comment: Line 618 “thus being better more efficient at suppressing…”. Please remove the word ‘better’.

Respone: This has been corrected.

  1. Comment: Lines 745-747 “Among all tumors, the mean levels of…”: What are the authors referring to as ‘all tumors’? Are these primary LGGs or primary and recurrent LGGs combined? Please, clarify.

Response: This section was intended to demonstrate that in LGGs, our analyses suggested a role for Mxd3 more consistent with an oncogene-like function rather than a TS-like function. In an effort to provide better focus and organization, particularly in response to Reviewer 2’s comments, we have decided to eliminate this paragraph in the revised manuscript since it does not directly address the role of Mxd3 as a tumor suppressor.

  1. Comment: Lines 824-828 “Mnt has slightly different sequence-specific DNA binding preferences compared to Myc and Mxd1-4 [26,281]. Like them however, it contains an mSin3 A/B-interacting SID domain that is necessary for the HDAC-mediated transcriptional repression of E-box containing target genes and the inhibition of MYC+RAS-mediated transformation.” The underlined part of the text only refers to the features of Mxd1-4 but not MYC. However, the phrase ‘Like them’ includes all MYC and Mxd1-4. Please, correct.

Response: This has been corrected and the ambiguity resolved.

  1. Comment: Lines 859-860 “Interestingly, rather than suppressing proliferation like Mxd1-4 or as it does in fibroblasts [26], Mnt was shown…”. According to the section on Mxd3, it does not suppress proliferation but promotes it instead. Therefore, please correct.

Response: We are unable to locate the reference to Mnt’s promotion of proliferation in the Mxd3 section.

20/21. (Considered together). Comment: Lines 919-948: These two paragraphs compare and contrast the role of Mnt and Myc in two different models. In addition to being distinct cancer models, the first one investigates the effect of heterozygous Mnt knockout whereas the second one looks at homozygous knockout. Could the authors please discuss the potential effect of these differences on the outcomes of these studies?

Comment:  Lines 963-977: Based on the findings summarized in this section, it seems that Mnt LOH (sometimes alongside MYC amplification) was pro-tumorigenic, implying Mnt’s role as a TSG in this context. However, in the 2 paragraphs in lines 910-937, the opposite seems to be the case despite the fact that they all look at T-cell-based models. How do the authors explain this contradiction? Please discuss in the text.

Response to Comments 20 and 21: The Reviewer correctly points out the seemingly contradictory findings between the consequences of Mnt knockout in mice, where, in collaboration with a MycT58A transgene, it seemingly serves as a pro-oncogenic factor, whereas In Sezary syndrome, it functions more as a classical tumor suppressor. Indeed, the findings in Sezary syndrome are recapitulated in CLL (a B-cell disorder) discussed immediately afterward. There are a number of non-mutually exclusive possibilities to explain this. For example, the murine studies were conducted in the background of an over-expressed/dysregulated pro-oncogenic MycT58A transgene whereas Sezary syndrome and CLL lymphocytes would not have carried this mutation, which tends to be associated with disease recurrence, particularly Burkitt’s lymphoma, and is rare even there. In the murine model, the expected TS activity of Mnt might have been lost or over-ridden as a result of Myc-driven lineage switching or higher level myeloid expression within the hematopoietic cell environment. Alternatively (and quite likely), MycT58A altered the expression of a subset of Myc targets (so-called “non-physiologic targets’) simply by virtue of its over-expression. Therefore, the seemingly contradictory murine and human findings are not necessarily incompatible. Rather they are entirely consistent with those shown in Fig. 2 and 3 that Mnt (as well as other members of the Extended Myc Network) can function as both oncogenes or TSGs in association with both favorable and unfavorable outcomes, an explanation that we reiterate multiple times throughout the review. In our revision, we have addressed this seeming paradox at the end of the section that discusses Mnt loss in Sezary syndrome.  

  1. Comment: 1001-1003 “Collectively, these studies suggest that the higher expression of Mnt associated with EMT in breast cancer actually serves to temper proliferation and thus serves as a TS.” : While Mnt serving as a TS in the context of proliferation is correct, this does not apply to the context of EMT, in which case, it serves more as a pro-tumorigenic factor. Therefore, could the authors please weigh and comment on both the aspects instead of being restricted to proliferation alone?

Response: At the end of the referred-to section we have added an explanation of how Mnt could also serve a potentially pro-oncogenic role by driving the EMT (“Nevertheless, it is still important to consider the potentially pro-oncogenic consequences of Mnt over-expression, which would tend to favor EMT both in breast and other cancers where it often contributes to such unfavorable behaviors as metastatic dissemination and chemotherapy resistance”).

  1. Comment: Line 1255 “However, the interaction between MCR IV and 14-3-3 requires neither glucose…” : In line 1248, the authors state that 14-3-3 interacts with MCR III instead. Which one is correct? Also, please be consistent between roman and Arabic numbers when mentioning MCRs.

Response: The statements are correct as written.  In the absence of glucose transcriptional control is more dependent on MCR IV whereas in glucose’s presence, the dependency switches to MCRs I-III.

     Arabic numbering has been changes to Roman.

  1. Comment: The paragraph begins by stating that “high levels of MondoA expression were found in leukemic blasts…”, implying its role as an oncogene. However, in lines 1357-1359, the authors refer to a study that showed that knockdown of MondoA in primary ALL blasts resulted in downregulation of genes associated with better therapeutic outcomes and survival. Therefore, it seems that despite being oncogenic, MondoA still confers a survival and therapeutic benefit. Could the authors please comment on that?

Response: The finding that the high level (pro-oncogenic) expression of Mondo A would be associated with a favorable outcome is not terribly surprising as there are many examples of this, not only among Extended Myc Network members (Fig. 3) but among other oncogenes as well, i.e. KRAS in renal cell cancer, MYB in gastric cancer, PI3C2B in urothelial cancer and many others (https://www.proteinatlas.org/).

  1. Comment: Lines 1449-1451 “It is tempting to consider that the loss of Mlxexpression whether by mutation, gene deletion or promoter silencing could contribute to the maintenance of the undifferentiated state and thus serve as a TS.” From this statement, it is unclear if Mlx itself or its loss of expression serves as a TS. Please rephrase to clarify.

Response: This was an admittedly awkward statement that has been modified: “It is tempting to consider that the loss of Mlx expression whether by mutation, gene deletion or promoter silencing inactivates its TS-like function and tips the balance in favor of the Myc Network, thereby maintaining both proliferation and the undifferentiated state.”

  1. Comment: Lines 1451-1475: The findings summarized in these paragraphs point to a pro-proliferative role of Mlx in normal hepatocytes. However, the findings in lines 1491-1510 suggest a tumor suppressive function. How do the authors explain these contrasting observations? Please comment.

Response: In a paper that has been submitted for publication and that is discussed in the above-referenced section of the review  (https://www.biorxiv.org/content/10.1101/2021.08.05.455215v2), we demonstrated that Mlx has both growth/tumor-promoting as well as TS-like functions that are context-dependent and consistent with the dual function of many if not all members of the Extended Myc Network as discussed throughout the review as well as above. Recurrent MLX gene deletions are associated with at least eight human cancer types (Fig. 2) and supported our finding that its deletion in hepatocytes leads to the development of hepatic adenomatosis (https://portal.gdc.cancer.gov/genes/ENSG00000108788). Interestingly, the genetic suppressors of hepatic adenomas and other benign tumors such as meningiomas, neurofibromas and uterine fibroids are very distinct from their more notorious counterparts such as TP53, RB, PTEN, APC and BRCA1/2 that suppress malignant tumors (Bluteau O, et al. Nat Genet 2002.32:312-5; Lee S, et al. Cancers (Basel) 2019.11(11); Navarro A, et al. PLoS One 2012.7(3):e33284. Williams EA, et al. Acta Neuropathol 2020.140:89-93.). In a new paragraph at the end of the Mlx section, we summarize this apparent paradox and draw parallels with the Myc Network.

  1. Comment: A majority of the Greek letters are missing from the text. Instead, there is just a blank space where the symbols should be (for example, TGF-b, HIF-a). Please, revise and correct.

Response: Corrections have been made to the deletion of the Greeks symbols which occurred during post-submission formatting of the article by the Cells editorial office.

  1. Comment: There are a few typos throughout the text. Please, revise and correct.

Response: We have corrected nearly a dozen such typos

Reviewer 2 Report

The review by Prochownik and Wang is a comprehensive review of the extended MYC network. While the review does an excellent job of presenting the studies that have emerged in this area, there is a significant lack of direction and conclusions. Overall, the review is extremely dense. A major restructuring would help to clarify the important take away messages and conclusion.

General Comments:
  • Article is very dense and could benefit from more structured sections for each transcription factor described. Ex. TS functions and studies, Oncogenic functions and studies, concluding remarks and gaps in knowledge.
  • Some sections end abruptly without proper conclusion to tie up section discussion.
  • Introduction and explanation of the review structure happens late in the review and should be discussed earlier.
  • Many experiments are discussed in great detail. It can detract from the overall point at which the author is making. It may benefit the author to condense experiments and more explicitly state conclusions and outcomes in the context of the section.
  • Title of paper seems misleading, this paper talks about growth suppression and acceleration outcomes within each transcription member in the extended Myc transcription network. Not all outcomes are growth suppressive.
  • There are some places within the text that 6-10 articles are cited, it may be better to cite a recent review that covers the topic instead.
  • Avoid using words like “they/them”, or “ones”. Due to the in-depth nature of the review, there are many different players mentioned, the author would benefit from stating them explicitly for clarity.

Author Response

Reviewer 2

  1. Comment: Article is very dense and could benefit from more structured sections for each transcription factor described. Ex. TS functions and studies, oncogenic functions and studies, concluding remarks and gaps in knowledge.

Response: We have shortened the manuscript somewhat and removed/rewritten several sections that were confusing or more tangential (for example see our responses to Reviewer 1’s comments 9,11,13,15 and 17). In actuality, each of the 13 sections in the original manuscript averaged only 114 lines. We gave serious consideration to providing additional reorganization in the manner suggested (i.e. having separate sub-sections as recommended above).  However, it is very difficult to do this without breaking up individual stories, jumping back and forth amongst them and causing confusion.   

  1. Comment: Some sections end abruptly without proper conclusion to tie up section discussion.

Response: We have now provided appropriate conclusions to those sections for which this was not previously done (Max, Mxd1, Mxd2, and Mlx).  

  1. Comment: Introduction and explanation of the review structure happens late in the review and should be discussed earlier.

Response: This was done deliberately. Since the focus of the review was the tumor suppressor-like functions of The Extended Myc Network, we crafted our Introduction from a historical perspective that initially focused on the original transforming function Myc, which was discovered before any other functions or players were known. We felt that doing this provided the reader with the context needed for understanding the roles of Extended Myc Network members in TS.

  1. Comment: Many experiments are discussed in great detail. It can detract from the overall point at which the author is making. It may benefit the author to condense experiments and more explicitly state conclusions and outcomes in the context of the section.

Response: See our responses to Comment 1 above. We have shortened/eliminated some of the unnecessary details of experiments that added little to the main points being made.

  1. Comment: Title of paper seems misleading, this paper talks about growth suppression and acceleration outcomes within each transcription member in the extended Myc transcription network. Not all outcomes are growth suppressive.

Response: From the time of the discovery of Myc-encoding retroviruses in the 1960s until the early 1990s, the Myc field focused on the role of the oncoprotein (either v-Myc or c-Myc) in maintaining either normal and neoplastic growth or the undifferentiated state. It was then shown that Max over-expression in vitro could suppress Myc’s activities, but only after the discovery of Mxd members was it appreciated that growth suppression was also a natural function of the Network (Fig. 1). A recurrent theme of our review is that some members can perform multiple and seemingly opposite functions. Our purpose was intended to emphasize the growth-suppressive properties of the Extended Myc Network, both in normal and neoplastic cells and tissues since few such reviews exist. However, the review would be incomplete, confusing, historically inaccurate and disingenuous if it addressed only these suppressive functions without proper reference to the better-known pro-oncogenic ones. Thus we do not feel that the inclusion of such background necessitates a change in title since the focus remains on growth suppression.  

  1. Comment: There are some places within the text that 6-10 articles are cited, it may be better to cite a recent review that covers the topic instead.

Response: In most cases, these articles were cited for other purposes as well. Because we wanted this review to be comprensive, we wanted to acknowledge the contributions of as many of the relevant players as possible.  We have also been criticized in past reviews for citing other reviews as sources.

Round 2

Reviewer 2 Report

The authors have address my comments sufficiently.